# On the Noise Robustness of In-Context Learning for Text Generation

**Hongfu Gao[1,2]\*, Feipeng Zhang[2], Wenyu Jiang[1,3], Jun Shu[4], Feng Zheng[5], Hongxin Wei[1]†**

[1]Department of Statistics and Data Science, Southern University of Science and Technology
[2]School of Economics and Finance, Xi'an Jiaotong University
[3]National Key Laboratory for Novel Software Technology, Nanjing University
[4]School of Mathematics and Statistics, Xi'an Jiaotong University
[5]Department of Computer Science and Engineering, Southern University of Science and Technology

## Abstract

Large language models (LLMs) have shown impressive performance on downstream tasks by in-context learning (ICL), which heavily relies on the quality of demonstrations selected from a large set of annotated examples. Recent works claim that in-context learning is robust to noisy demonstrations in text classification. In this work, we show that, on text generation tasks, noisy annotations significantly hurt the performance of in-context learning. To circumvent the issue, we propose a simple and effective approach called Local Perplexity Ranking (LPR), which replaces the "noisy" candidates with their nearest neighbors that are more likely to be clean. Our method is motivated by analyzing the perplexity deviation caused by noisy labels and decomposing perplexity into inherent perplexity and matching perplexity. Our key idea behind LPR is thus to decouple the matching perplexity by performing the ranking among the neighbors in semantic space. Our approach can prevent the selected demonstrations from including mismatched input-label pairs while preserving the effectiveness of the original selection methods. Extensive experiments demonstrate the effectiveness of LPR, improving the EM score by up to 18.75 on common benchmarks with noisy annotations. Our code is available at https://github.com/ml-stat-Sustech/Local-Perplexity-Ranking

## 1 Introduction

Large language models (LLMs) have shown remarkable performance on downstream tasks by *in-context learning* (ICL) with only a few task demonstrations [7, 10]. Without requiring explicit parameter updates, in-context learning consistently outperforms zero-shot inference on various tasks (e.g., classification and generation), making it a compelling alternative to supervised fine-tuning [13, 16]. In particular, the success of ICL heavily relies on the quality of demonstrations selected from a large set of annotated examples [21, 29, 51, 60]. For those candidates, input-label mappings solicited from humans [61, 73] or LLMs [58] can often be noisy, especially in complex tasks. This gives rise to the importance of *noise-robust ICL*, which aims to construct effective demonstrations in the presence of noisy and erroneous labels.

Previous works show that in-context learning on classification tasks is fairly robust to label noise in the in-context demonstrations [9, 12, 32, 37, 54, 55]. However, it is still mysterious how noisy labels affect the performance of ICL on text generation tasks. In this work, we present the first study on in-context learning with a *noisy* annotated dataset for generation. Surprisingly, we empirically find that label noise in the demonstrations significantly degrades ICL's performance on generation tasks,

---

\*Work done while working at SUSTech as a visiting scholar.
†Corresponding author (`weihx@sustech.edu.cn`)

which is different from previous results on classification. Moreover, increasing the number of selected demonstrations with a fixed noise rate or utilizing more effective selection methods (e.g., TopK [28] and DPP [62]) will intensify the negative effect of noisy labels. This motivates our method, which can universally improve the noise robustness of existing selection methods for in-context learning.

In this paper, we show that the issue of noisy annotations can be mitigated through the perplexity ranking of noisy candidates (i.e., input-label pairs) during selection. Our method, Local Perplexity Ranking (dubbed **LPR**), is motivated by our analysis of the perplexity deviation caused by noisy labels (i.e., incorrect answers). We find that wrong answers generally result in a higher perplexity of large language models compared to correct ones, in response to the same question. To explain this phenomenon, we decompose the perplexity into two components: inherent perplexity, which measures the task complexity of the question and the correct answer, and matching perplexity, which assesses the perplexity deviation caused by noisy outputs.

Therefore, our key idea behind Local Perplexity Ranking is to decouple the matching perplexity by performing the ranking among the neighbors in semantic space. This can be achieved by ranking candidates' perplexity alongside their nearest neighborhoods, which usually have similar levels of inherent perplexity. In particular, we replace each low-rank candidate selected by existing methods (e.g., random, TopK, and DPP) with its nearest neighbor that is highly ranked. In effect, our LPR strategy can prevent the selected demonstrations from containing mismatched input-label pairs while preserving the effectiveness of the original selection methods. In this way, we ensure the correctness and relevancy of demonstrations, thereby improving the noise-tolerant ability of in-context learning.

To verify the effectiveness of our method, we conduct extensive evaluations on six text generation datasets, including NQ [22], WebQ [5], SQuAD [46], SCIQ [56], GeoQuery [39] and NL2Bash [27] datasets. The results demonstrate that local perplexity ranking can largely improve the noise-robustness of all existing selection methods under irrelevant and relevant noises. For example, on SCIQ with $60\%$ irrelevant label noise, LPR improves the exact match score of the TopK method from 29.31 to 48.06 – a significant direct improvement of **18.75**. Moreover, our method can be easily adopted in practice. The performance of LPR is insensitive to the hyperparameters, including the threshold $\gamma$ and the number of local neighbors $k$. This approach can effectively generalize to various LLMs to improve their noise-robustness with in-context learning.

Our contributions are summarized as follows:

- We present the first study to show that annotation quality is crucial for in-context learning in text generation, where noisy annotations significantly hurt the performance. Increasing the set size of demonstrations cannot bridge the gap, as well as picking other selection methods.

- We propose Local Perplexity Ranking (LPR), a simple and effective method to enhance the noise robustness of in-context learning. The key idea is to decouple the matching perplexity by performing the ranking among the neighbors of each candidate in semantic space.

- We empirically show that LPR can improve the noise robustness of existing demonstration selection methods in ICL across various types of label noise. In addition to text generation, we also validate the effectiveness of our method in text classification tasks.

## 2 Preliminary

### 2.1 In-context learning for generation

We consider in-context learning (ICL) of large language models (LLMs) in generation tasks, where we aim to generate text outputs $\boldsymbol{y} = (y_1, ..., y_{|\boldsymbol{y}|})$ (i.e., token sequences) conditioned on the inputs $\boldsymbol{x} = (x_1, ..., x_{|\boldsymbol{x}|})$ and the context $\boldsymbol{C}_K$. In particular, the context $\boldsymbol{C}_K = \{(\boldsymbol{x}_i, \boldsymbol{y}_i)\}_{i=1}^K$ contains $K$ task demonstrations (e.g., input-output pairs), selected from a large annotated dataset with $N$ examples $\mathcal{D} = \{(\boldsymbol{x}_j, \boldsymbol{y}_j)\}_{j=1}^N$. Given a new test input text $\boldsymbol{x}_{test}$, we make the generation of output $\boldsymbol{y}_{test}$ via large language models as

$$\boldsymbol{y}_{test} \sim \mathcal{P}_{LLM}(\boldsymbol{y}_{test} \mid \{(\boldsymbol{x}_i, \boldsymbol{y}_i)\}_{i=1}^K, \boldsymbol{x}_{test}), \tag{1}$$

where $\sim$ refers to decoding strategies(e.g. greedy decoding and nuclear sampling [17, 62]). Generation with the ICL procedure is especially attractive as it does not require the parameter updating of large language models, which is often expensive and impractical.

Table 1: An illustration of the effect of three different types of annotated dataset for in-context learning. The middle column is in-context demonstrations, and the last column is the Llama2-7B [49] model prediction. The model tends to learn the label of the demonstration.

| Test Input | **Support**: All forms of life are built of at least one cell. A cell is the basic unit of the structure and function of living things.
**Question**: What are the smallest structural and functional units of all living organisms?
**Output**: | |
|---|---|---|
| **Setting** | **In-Context Demonstration** | **Prediction** |
| Clean | **Support**: Cells are organized into tissues, tissues are organized into organs.
**Question**: What is considered the smallest unit of the organ?
**Output**: Cells | Cells |
| Irrelevant | **Support**: Cells are organized into tissues, tissues are organized into organs.
**Question**: What is considered the smallest unit of the organ?
**Output**: Earth | Earth |
| Relevant | **Support**: Cells are organized into tissues, tissues are organized into organs.
**Question**: What is considered the smallest unit of the organ?
**Output**: tissues | tissues |

Existing studies show that the selection strategy of demonstration plays a crucial role in the ICL performance [25, 31, 43, 44, 47]. A naive method is to randomly sample the demonstrations from annotated examples without repetition [36]. To introduce the relevancy, TopK [28] proposes to select the closest examples to the test input in the embedding space

$$\boldsymbol{C}_K = \mathcal{R}_K(\boldsymbol{x}_{test}) = \mathrm{TopK}_{\boldsymbol{x}}(s(\boldsymbol{x}_{test}, \boldsymbol{x})),$$

where $\mathcal{R}$ is a retriever, $s(\boldsymbol{x}_{test}, \boldsymbol{x})$ denotes the cosine similarity score between $\boldsymbol{x}_{test}$ and examples $\boldsymbol{x}$ from the annotated dataset. We use $\mathrm{TopK}$ to denote the top $K$ examples ranked by the score.

These selection strategies focus on the inputs of demonstrations, assuming that all examples are labeled correctly in the large dataset [28, 36, 62]. However, collecting a large-scale dataset with perfectly correct labels is challenging and expensive, especially for generation tasks [2, 64]. In practice, researchers often use crowdsourcing [61, 73] or large language models (LLMs) [58] such as GPT-4 [38] to create input-output pairs for new tasks, which inevitably leads to some mistakes in the annotations. This motivates us to analyze the issue of label quality in ICL for generation tasks.

## 2.2 Setting of noisy annotations

Given a large-scale dataset with noisy annotations $\tilde{\mathcal{D}} = \{(\boldsymbol{x}_j, \tilde{\mathbf{y}}_j)\}_{j=1}^N$, the selected demonstration might contain mismatched input-output pairs $(\boldsymbol{x}, \tilde{\mathbf{y}})$, i.e., the output $\tilde{\mathbf{y}}$ might be not a correct answer to the input $\boldsymbol{x}$. Conditioned on the noisy demonstrations, the generation of output via ICL is made as

$$\mathbf{y}_{test} \sim \mathcal{P}_{LLM}(\mathbf{y}_{test} \mid \{(\boldsymbol{x}_i, \tilde{\mathbf{y}}_i)\}_{i=1}^K, \boldsymbol{x}_{test}). \tag{2}$$

In the real world, noisy annotations may arise from unintentional mistakes or limited knowledge, resulting in various types of noise in the demonstrations. In this work, we define two categories of noisy annotations based on the input-output relevance, as follows:

**Irrelevant noise** assumes that the generation of noisy annotations is conditionally independent of inputs. For example, crowdsource workers may make mistakes accidentally, introducing random words or sentences in annotations. This can be simulated by reconstructing the output with random words from a subset that does not contain tokens presented in the original input-output pairs.

**Relevant noise** is a more realistic setting where the corrupted output is relevant to the inputs despite its incorrectness. This type of corruption may occur due to the limited knowledge of annotators and LLMs. We simulate the relevant noise by generating related yet incorrect outputs using ChatGPT-4.

In Table 1, we present an ICL example of question answering (QA) tasks to illustrate the difference between the two noisy settings. In this example, the clean annotation for the test input is "Cells". For noisy annotations, the irrelevant noise is randomly sampled as "Earth", while the relevant noise "tissues" exists in the support of in-context demonstration. We proceed by analyzing the empirical effects of noisy annotations in generation tasks.

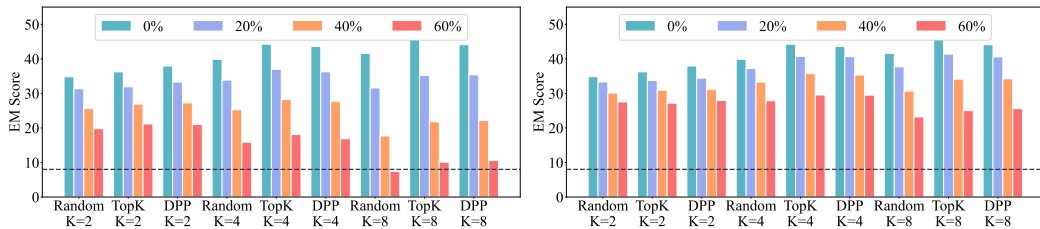

Figure 1: Average ICL performance with noisy annotations in various generation tasks across different demonstration settings. Both the two types of noises significantly deteriorate the performance of in-context learning on text generation tasks. The black line denotes zero-shot performance.

## 3 Empirical study of noisy ICL in text generation

In this section, we investigate the impact of noisy annotations on the performance of in-context learning for text generation. In particular, we conduct experiments on three types of generation tasks, including: question answering (NQ [22], WebQ [5]), reading comprehension (SQuAD [46], SCIQ [56]), code generation (GeoQuery [39], NL2Bash [27]). To simulate the noise, we generate noisy annotations with a pre-defined probability (e.g., 20%, 40%, 60%) in the annotated datasets. We use the output of an input from a different generation task as irrelevant noise, and adopt ChatGPT-4 to generate relevant yet false outputs as relevant noise. Furthermore, we compare the performance of noisy ICL with demonstrations across various set sizes (e.g., 2, 4, 8) and selection methods, including Random [36], TopK [28] and DPP [62]. Following previous work [16, 28, 62], we report the average Exact Match (EM) score with Llama2-7B [49].

**ICL is not robust to noisy annotations in text generation.** Figure 1 presents the empirical results of ICL methods with noisy annotations. The results show that both the two types of noises significantly deteriorate the performance of in-context learning on text generation tasks, which is different from the observations of ICL on classification tasks [9, 12, 32, 37, 54, 55]. In particular, a higher noise rate in annotated datasets leads to poorer performance of in-context learning. Moreover, irrelevant noises have a more negative influence than relevant noises, which may benefit the inference in the way of *task recognition* [40].

**The impact of demonstration selection.** To provide a deep understanding of noisy annotations, we analyze the performance of noisy ICL across different demonstration settings, including the set size (i.e., $K$) and selection methods. Results in Figure 1 show that, under the noisy settings, selecting a larger set of demonstrations does not enhance — and may even worsen — the performance of text generation. For example, the ICL performances with $K = 8$ are basically lower than those with $K = 2$, which is inconsistent with the clean setting. In addition, the advantages of those powerful selection methods (i.e., TopK and DPP) are neutralized in the presence of noisy annotations.

Through the empirical analysis, we find that noisy annotations significantly hurt the performance of ICL in text generation tasks. More importantly, increasing the set size of demonstrations cannot bridge the gap, as well as picking an existing selection method, like DPP. This motivates us to design *noise-robust* methods, which can universally improve the noise robustness of in-context learning.

## 4 Methodology

In this section, we first analyze the perplexity deviation caused by noisy annotations and introduce the disentanglement of perplexity to explain the phenomenon. In light of this, we propose a novel method – local perplexity ranking – to improve the noise robustness of in-context learning for text generation. Our method can be easily incorporated into existing methods of demonstration selection.

### 4.1 Perplexity deviation of noisy annotations

For language models, perplexity measures the degree of uncertainty in generating new tokens. In particular, a low perplexity indicates that the model makes the prediction with high confidence.

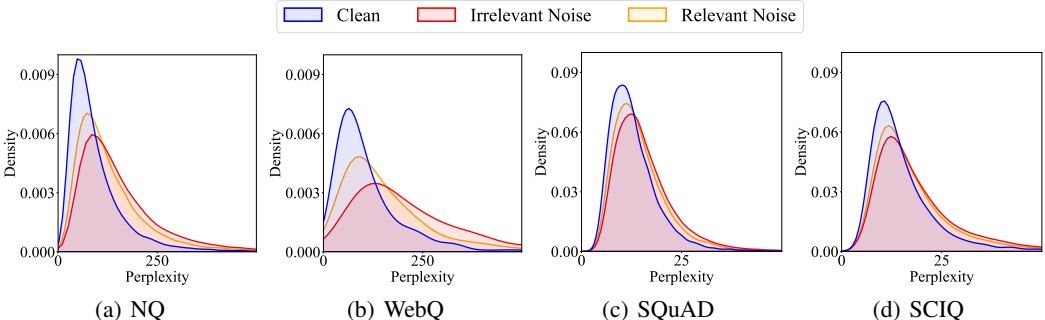

Figure 2: The distribution of perplexity of Llama2-7B [49] on clean and noisy annotations. Examples with noisy annotations indeed obtain higher perplexity than those with clean annotations.

Therefore, perplexity is commonly used to evaluate the language quality of generated content, e.g., detecting attack prompts [3], out-of-distribution instances [4, 57], hard-to-learn instances [13], and corrupted instances [64]. In light of this, we conjecture that mismatched input-output pairs may result in higher perplexity of LLMs due to their low co-occurrence rate. For instance, in the example presented in Table 1, the term "earth" rarely co-occurs with "cells" and "organ", so LLMs are more likely to exhibit high perplexity in the input-output pair.

**Empirical study**  To validate this assumption, we compare the perplexity of clean and noisy annotations in text generation tasks. Specifically, we concatenate each tokenized input-output pair $(\boldsymbol{x}, \boldsymbol{y})$, and obtain the corresponding tokenized sequence $\boldsymbol{z} = (z_1, ..., z_{|\boldsymbol{z}|}) = (x_1, ..., x_{|\boldsymbol{x}|}, y_1, ..., y_{|\boldsymbol{y}|})$, where $|\boldsymbol{z}| = |\boldsymbol{x}| + |\boldsymbol{y}|$. Now, the perplexity of $\boldsymbol{z}$ is calculated as:

$$\text{Perplexity}(\boldsymbol{z}) = \exp\{-\frac{1}{|\boldsymbol{z}|} \sum_{i=1}^{|\boldsymbol{z}|} \log p_\theta(z_i|z_{<i})\}, \tag{3}$$

where $\log p_\theta(z_i|z_{<i})$ is the log-likelihood of the $i$-th token conditioned on the preceding tokens $z_{<i}$, from the given language model parameterized by $\theta$.

In Figure 2, we present the perplexity distribution of Llama2-7B [49] on clean and noisy annotations of four datasets. The results illustrate that examples with noisy annotations indeed obtain higher perplexity than those with clean annotations, which confirms our assumption. In particular, relevant noises achieve slightly lower perplexity than irrelevant noises since relevant outputs are close to the inputs despite their erroneous information. However, the deviation of the perplexity distribution caused by noisy annotations is marginal, making it suboptimal to differentiate noisy annotations from clean ones. In the following, we explain this phenomenon with the disentanglement of perplexity.

**Disentanglement of perplexity**  Given an input-output pair, the perplexity of large language models (LLMs) stems not only from how well the output matches the input, but also from the inherent complexity of the task. For example, a mathematical question with a correct answer can exhibit a higher perplexity than a question of daily life with an incorrect answer. Informally, we decompose the overall Perplexity into two components [3], as shown below:

$$\text{Perplexity} = \text{Inherent Perplexity} + \text{Matching Perplexity}$$

Here, the inherent perplexity measures how the model is familiar with the task (i.e., the input and the correct output). The matching perplexity quantifies the perplexity deviation caused by noisy outputs, so it can be zero with correct outputs. A higher matching perplexity indicates that the output is more likely to be incorrect for the input. However, directly computing the matching perplexity is non-trivial as clean outputs are unknown. To circumvent the issue, we aim to design an effective method to decouple the matching perplexity from the overall perplexity.

---

[3]This disentanglement is conceptual rather than mathematical.

## 4.2 Local Perplexity Ranking

**Intuition**   Motivated by the previous analysis, we propose *local perplexity ranking* (LPR), a general strategy that can improve the noise robustness of in-context learning. Our key idea is to decouple the matching perplexity by performing the ranking among the neighbors in semantic space. Here, our approach is built on two natural assumptions that are naturally satisfied in the real world:

1. The clean annotations are the majority in the annotated dataset.
2. Examples that are semantically similar share the same level of inherent perplexity.

In the literature, Assumption 2 is also supported by previous findings that paragraphs whose representations are close to each other share the same intrinsic task [14, 28, 73]. With the two assumptions, we can approximate the inherent perplexity of a candidate through its neighbors, where most examples are correctly annotated. In other words, the candidate is more likely to be wrongly annotated if its perplexity is relatively higher than its neighbors, and vice versa. With this in mind, we present the details of our approach in the following.

**Finding the local neighbors**   Given a test input, we first sample a candidate set $\widetilde{C}$ with a pre-defined selection strategy, such as Random [36], TopK [28] or DPP [62]. For each candidate $z^*$, we adopt $k$-Nearest-Neighbors ($k$-NN) to find its local neighbors that are close to the candidate in token space. Formally, the $k$ local neighbors are obtained as: $N_k(z^*) = \{z_{\pi(1)}, z_{\pi(2)}, ..., z_{\pi(k)}\}$, where $\pi(i)$ is the index of the example with the $i$-th smallest distance to the candidate. In particular, we use the cosine similarity score to measure the distance between the candidate $z^*$ and other examples $z$:

$$\cos(z_i, z^*) = \frac{z_i^\top z^*}{||z_i||_2 ||z^*||_2}.$$

**Ranking the perplexity**   As discussed above, the local neighbors share the same level of inherent perplexity, which enables the comparison of their matching perplexity. For each candidate $z^*$, we propose to rank the perplexity of examples in the cluster of local neighbors $z^* \cup N_k(z^*)$. Formally, we first sort all examples in the cluster in increasing order by the perplexity and obtain the original indices for the sorted scores as:

$$\mathcal{I} = \operatorname{argsort}\{\operatorname{Perplexity}(z_n)\}_{n=1}^{k+1}, \quad z_n \in (z^* \cup N_k(z^*)), \tag{4}$$

where $\operatorname{Perplexity}(\cdot)$ is the overall perplexity defined in Equation 3. In this way, the high-ranking examples are more likely to be correctly annotated than the low-ranking example in the sorted list $\mathcal{I}$.

**Substituting the noisy candidates**   To build the final demonstration set, we propose to replace the noisy candidates with their nearest neighbors that are more likely to be clean. In particular, we can determine whether a candidate should be replaced by:

$$g(z_n) = \mathbb{1}\left(\frac{\operatorname{Loc}(z_n, \mathcal{I})}{k+1} \geq \gamma\right), \tag{5}$$

where $\gamma$ is the pre-defined threshold (e.g., 50%), $\mathbb{1}(\cdot)$ is the indicator function and $\operatorname{Loc}(z_n, \mathcal{I})$ return the index of $z_n$ in the sorted list $\mathcal{I}$. It is worth noting that the proposed method is not sensitive to the value of the hyperparameter $\gamma$, as shown in Subsection 5.1. Then, for those candidates with $g(z_n)$, we pick the substitutes from their neighbors by:

$$\min\{i \in N^k | g(z_{\pi(i)}) = 0\},$$

where $\pi(i)$ is the index of the example with the $i$-th smallest distance to the candidate. After the replacement, we establish the final demonstration set for in-context learning. Noticeably, our method offers several compelling advantages:

- **Algorithm-agnostic**: LPR can be easily incorporated into existing demonstration selection methods, consistently improving the robustness against noisy annotations.
- **Easy to use**: LPR does not require heavy hyperparameter tuning, as it is insensitive to the threshold value (see Figure 3). LPR does not introduce much computational cost due to the efficient computation of perplexity (see Table 4).

Table 2: Main results on various datasets. The bold indicates the improvement by integrating LPR.

| Dataset | Method | Clean 0% | Irelevant Noise 20% | 40% | 60% | Relevant Noise 20% | 40% | 60% |
|---------|--------|----------|---------------------|-----|-----|--------------------|-----|-----|
| NQ | Random | 14.51±0.51 | 10.97±0.29 | 7.37±0.45 | 4.23±0.46 | 12.00±0.65 | 9.67±0.45 | 6.40±1.02 |
| | **+Ours** | **15.05±0.10** | **13.31±0.25** | **11.51±0.51** | **8.87±0.74** | **13.74±0.12** | **13.28±0.33** | **9.43±0.52** |
| | TopK | 20.25±0.10 | 13.95±1.14 | 9.97±1.13 | 5.90±1.08 | 16.21±0.22 | 12.22±0.22 | 8.50±0.28 |
| | **+Ours** | 19.19±0.19 | **17.15±0.50** | **13.54±0.41** | **9.64±0.25** | **17.25±0.69** | **14.82±0.51** | **11.98±0.60** |
| | DPP | 20.35±0.76 | 14.69±0.94 | 9.87±0.49 | 5.97±0.48 | 15.47±1.00 | 11.28±0.42 | 7.89±0.25 |
| | **+Ours** | 19.68±0.33 | **16.59±0.45** | **13.31±0.57** | **11.18±0.50** | **16.79±0.47** | **14.91±0.18** | **11.94±0.91** |
| WebQ | Random | 20.37±0.64 | 15.18±1.06 | 10.39±0.83 | 4.83±0.17 | 18.29±0.43 | 15.92±0.68 | 13.50±0.17 |
| | **+Ours** | **21.94±0.64** | **20.32±0.92** | **16.33±0.58** | **12.54±0.29** | **21.51±0.33** | **19.33±0.41** | **16.69±1.11** |
| | TopK | 30.16±0.58 | 22.52±0.64 | 14.52±0.78 | 8.00±1.12 | 27.19±0.27 | 22.82±0.75 | 18.88±1.09 |
| | **+Ours** | 29.24±0.34 | **26.55±0.24** | **21.67±1.28** | **14.54±1.02** | **28.49±0.43** | **25.44±0.68** | **21.28±0.12** |
| | DPP | 29.40±0.39 | 22.11±0.81 | 13.72±0.27 | 7.33±0.68 | 26.18±1.04 | 21.53±0.61 | 16.80±0.17 |
| | **+Ours** | **29.92±0.48** | **26.57±0.95** | **21.94±1.05** | **14.85±0.81** | **28.46±1.01** | **25.61±0.78** | **21.35±1.17** |
| SQuAD | Random | 56.50±0.57 | 50.00±0.62 | 39.10±0.88 | 26.20±0.79 | 53.90±0.65 | 49.17±0.62 | 42.03±0.79 |
| | **+Ours** | **57.73±0.79** | **56.87±0.47** | **48.50±0.86** | **43.00±0.86** | **57.70±1.31** | **53.93±0.33** | **47.93±0.48** |
| | TopK | 56.97±0.69 | 51.83±1.03 | 42.83±1.68 | 29.10±2.92 | 54.77±0.69 | 49.37±1.37 | 41.37±2.09 |
| | **+Ours** | **57.27±0.62** | **55.40±0.37** | **51.43±1.26** | **41.30±2.65** | **56.90±0.64** | **53.90±1.08** | **48.37±0.66** |
| | DPP | 57.29±0.87 | 50.57±0.33 | 41.63±1.00 | 25.67±2.52 | 56.10±0.59 | 49.57±1.24 | 43.37±0.78 |
| | **+Ours** | **58.10±0.29** | **56.73±0.61** | **52.53±0.33** | **42.93±0.88** | **57.50±0.54** | **55.90±0.18** | **50.77±0.39** |
| SCIQ | Random | 68.15±0.28 | 59.19±1.57 | 44.19±2.89 | 28.21±2.96 | 64.59±1.42 | 58.39±0.16 | 49.54±0.80 |
| | **+Ours** | 67.93±0.85 | **65.06±1.34** | **55.57±0.53** | **42.00±2.96** | **66.63±0.94** | **62.70±1.10** | **58.92±1.74** |
| | TopK | 68.62±1.13 | 59.59±1.28 | 45.77±2.68 | 29.31±1.73 | 64.66±1.34 | 58.54±0.12 | 49.47±0.65 |
| | **+Ours** | **70.06±0.32** | **66.67±0.81** | **57.44±1.04** | **48.06±1.53** | **67.76±0.50** | **63.96±1.71** | **56.32±2.18** |
| | DPP | 67.29±0.35 | 57.69±1.83 | 45.34±1.56 | 28.50±1.78 | 64.88±0.43 | 58.91±0.64 | 50.00±0.85 |
| | **+Ours** | **70.57±0.45** | **67.86±1.43** | **59.65±2.11** | **45.46±2.72** | **69.16±0.98** | **65.63±0.21** | **56.72±1.37** |
| GeoQuery | Random | 27.97±0.99 | 23.18±0.62 | 17.44±1.56 | 14.10±0.74 | 26.48±0.17 | 26.13±0.05 | 26.25±0.40 |
| | **+Ours** | 27.27±0.36 | **27.12±0.69** | **25.52±1.02** | **22.23±0.67** | **27.43±0.71** | **27.01±0.05** | **26.73±0.90** |
| | TopK | 44.17±0.09 | 27.28±2.65 | 17.49±2.05 | 9.96±3.08 | 41.31±0.46 | 38.48±0.63 | 34.90±0.69 |
| | **+Ours** | 43.32±0.05 | **42.25±1.00** | **33.80±1.43** | **24.39±1.08** | **42.59±0.37** | **39.40±0.37** | **37.74±1.23** |
| | DPP | 45.81±0.71 | 31.79±5.93 | 21.54±3.36 | 10.61±0.15 | 42.97±1.96 | 39.91±0.42 | 33.34±0.53 |
| | **+Ours** | 44.18±0.47 | **43.01±0.02** | **40.94±0.91** | **33.25±1.27** | 41.49±0.11 | **40.62±0.06** | **36.81±0.61** |
| NL2Bash | Random | 27.91±0.37 | 25.37±0.21 | 15.77±0.91 | 8.95±0.65 | 27.20±1.06 | 28.09±0.51 | 26.27±0.56 |
| | **+Ours** | **29.93±1.18** | **29.09±0.26** | **26.04±2.05** | **22.92±0.39** | **29.01±0.36** | **28.92±0.07** | **26.80±0.55** |
| | TopK | 35.71±0.42 | 27.40±0.26 | 20.00±0.62 | 9.95±0.68 | 32.57±0.13 | 30.21±0.08 | 27.48±0.35 |
| | **+Ours** | 33.92±0.70 | **32.51±1.59** | **30.50±1.02** | **23.47±1.52** | 31.33±0.04 | **31.39±1.70** | **29.49±0.06** |
| | DPP | 37.77±0.02 | 31.52±0.12 | 23.23±0.34 | 11.16±2.14 | 32.74±0.29 | 32.56±0.61 | 26.72±1.58 |
| | **+Ours** | 35.85±1.51 | **32.27±0.99** | **32.47±0.40** | **27.84±1.17** | **33.63±0.23** | 32.53±0.57 | **28.96±0.98** |

## 5 Experiments

### 5.1 Experimental Setup

**Datasets.** We employ 6 generation datasets for the evaluations, including **Open-Domain Question-Answering**: NQ [22], WebQ [5]; **Reading Comprehension**: SQuAD [46] and SCIQ [56]; **Code Generation**: GeoQuery [39] and NL2Bash [27]. Due to limited space, these tasks' input/output, statistics, split and evaluation metrics are reported in Appendix A.2.

**Models and ICL methods.** For the main results, we use Llama-2-7B-Chat [49] as the LLM throughout our experiments. We also provide experiments on other models including Llama2-13B-Chat [49], Mistral-7B [19] and OPT-6.7B [66]. We use bert-base-uncased sentence encoder as the similarity tokenizer [11, 62]. We conduct experiments with existing demonstration selection methods, including **Random** [36], **TopK** [28] and **DPP** [62]. For hyperparameters, we set the number of neighbors $k = 4$ and the threshold $\gamma = 50\%$ by default. The details of our implementation is presented in Appendix A.2.

### 5.2 Main Results

**Can LPR improve the noise-robustness of in-context learning?** Table 2 presents the average in-context learning performance of the baselines and our method on six generation tasks, under various types of noisy annotations. A salient observation is that our method drastically improves the the noise-robustness performance of the existing demonstration selection methods by employing LPR.

Table 3: Average test performance of the baselines and our method using varying large language models across various noise types. The results are shown as Naive/+Ours. The bold indicates the improved results by integrating LPR.

| Method | Clean | Irelevant Noise | | | Relevant Noise | | |
|--------|-------|-----------------|--|--|----------------|--|--|
| | 0% | 20% | 40% | 60% | 20% | 40% | 60% |
| Llama2-13B [49] | 45.13/**45.27** | 38.58/**43.47** | 29.00/**39.24** | 18.93/**30.46** | 42.18/**44.32** | 37.10/**41.88** | 30.67/**36.76** |
| Mistral-7B [19] | 34.89/34.12 | 32.12/**33.59** | 26.28/**31.56** | 19.24/**27.03** | 33.43/**33.91** | 30.52/**32.64** | 26.63/**30.00** |
| OPT-6.7B [66] | 23.46/**24.03** | 17.26/**21.31** | 11.32/**17.29** | 7.68/**12.91** | 20.16/**22.40** | 17.58/**20.22** | 14.95/**17.52** |

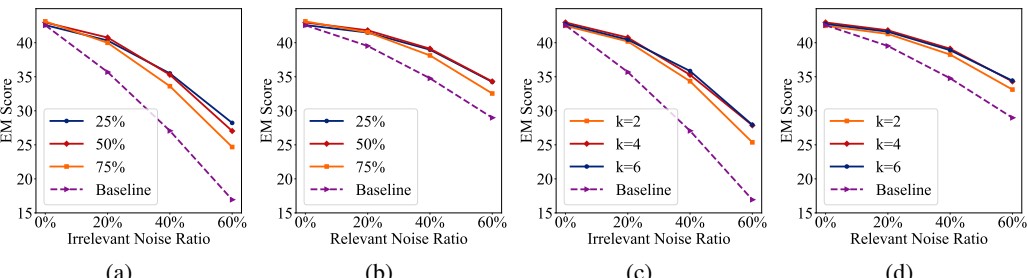

Figure 3: The average test performance with different thresholds $\tau$ and numbers of local neighbors $k$ across various noise types. Figure (a) and (b) analyze how the hyperparameter $\tau$ affects the performance of LPR. Figure (c) and (d) illustrate the influence of the hyperparameter $k$.

For example, on the NQ with $60\%$ irrelevant noise, our approach improves the EM score of the naive random selection method from 28.21 to 42.00 -a **13.79** of direct improvement. Moreover, we show that the LPR can boost performance for a wide range of existing demonstration selection methods such as TopK [28] and DPP [62]. For example, we observe that, on SCIQ with $60\%$ irrelevant label noise, LPR improves the exact match score of the TopK method from 29.31 to 48.06 – a significant direct improvement of **18.75**. Our method also establish strong robustness against all types of noisy annotations. Appendix A.3 reports the results with various demonstration sizes.

**How does the threshold $\gamma$ affect the noise-robustness of LPR?** In Figure 3 (a) and (b), we ablate how the parameter $\gamma$ in our method (cf. Eq. 5) affects the noise-robust performance. The base indicates all candidate demonstrations are selected without our method. It's noteworthy that LPR shows robustness to the choice of threshold $\gamma$, even if we set $\gamma = 75\%$ also yield significant EM score improvements. We can also observe that as the threshold $\gamma$ decrease, the noise-robust performance also improve, especially under $60\%$ noise conditions. Due to space constraints, we only report the average results of multiple baselines on various generation tasks.

**Does LPR work with the different number of $k$ nearest neighbors?** We evaluate how the number of nearest neighbors $k$ in our method affects the LPR performance. Specifically, We vary the number of neighbors $k = \{2, 4, 6\}$. As is shown in Figure 3 (c) and (d), an increase in the number of nearest neighbors beyond 0 leads to an evident improvement in EM score, and the performance starts to reach a point of saturation with the further addition of neighbors. Concernedly, more perplexity of nearest neighbors needs to be calculated as $k$ value increase, but the improvement is limited. For simplicity, we employ a moderate range of neighbors and use $k=4$ throughout our experiments.

**Is LPR effective with different LLMs?** To show our proposed method is model-agnostic, we conduct experiments on a diverse collection of model architectures and present the results in Table 3. From the results, we observe that our method consistently improves the ICL performance when using Llama2-13B [49], Mistral-7B [19] and OPT-6.7B [66]. For instance, with Mistral-7B, using our method boosts the ICL performance using the random selection method from 19.24 to 27.07, an average **7.83** of direct improvement on 6 datasets with irrelevant-60% noisy annotations.

## 6 Discussion

**Global Perplexity Ranking vs. Local Perplexity Ranking.** While our method has demonstrated strong promise in in-context learning, one may also ask: *can a similar effect be achieved by selecting*

Table 4: Average test performance comparison between global perplexity ranking and local perplexity ranking. The results are shown as *Global/Local*. Bold numbers are superior results.

| Method | Clean 0% | Irrelevant Noise 20% | 40% | 60% | Relevant Noise 20% | 40% | 60% | Time (h) |
|---|---|---|---|---|---|---|---|---|
| Random | 39.32/**40.66** | 38.94/38.89 | **34.41**/32.98 | 27.82/26.59 | 39.23/**39.90** | 36.38/**37.31** | 31.76/**33.24** | 2.88/**0.55** |
| TopK | 40.57/**43.94** | 39.94/**41.44** | 35.85/**36.02** | 31.79/28.38 | 40.33/**42.60** | 38.69/**39.53** | 33.88/**34.48** | 3.06/**0.57** |
| DPP | 42.33/**44.32** | 40.18/**41.94** | 36.20/**36.86** | 30.91/28.60 | 40.42/**42.98** | 38.49/**40.51** | 32.24/**35.20** | 3.21/**0.64** |
| Average | 40.74/**42.97** | 39.68/**40.76** | **35.49**/35.28 | 30.17/27.86 | 39.99/**41.83** | 37.85/**39.12** | 32.63/**34.31** | 3.05/**0.57** |

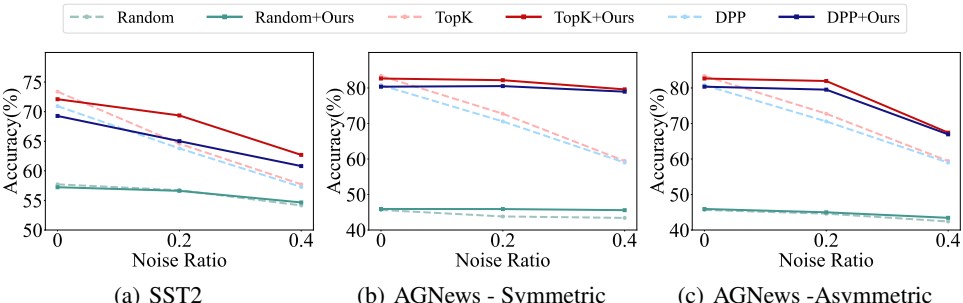

Figure 4: Average test accuracy on SST2 [48] and AGNews [67]. Different colors indicate the selection methods. The solid lines denote existing selection methods, and the dotted lines represent the method integrated by our method. We omit the noisy type on the binary classification – SST2.

*demonstrations with the lowest perplexity in the whole dataset?* In this ablation, we compare our method with a global perplexity ranking method that selects demonstrations with the lowest perplexity values of input-label pairs from a large candidate set (e.g., $\{(\boldsymbol{x}_i, \boldsymbol{y}_i)\}_{i=1}^{100}$).

Table 4 presents the performance comparison between our method and the global perplexity ranking method. While both the two perplexity ranking methods improve the robustness of ICL against noisy annotations, the global approach obtains inferior performance compared to our proposed method in most cases, especially in the cases of clean and low noise rates. In efficiency, Table 4 also show that the local ranking approach requires only 20% of the time required by the global ranking. This is because our method only calculates the perplexity of the local neighbors for each candidate, instead of using a large candidate pool. Overall, we show that the global ranking method cannot outperform the local ranking while introducing much more computational loads.

**Transfer to text classification tasks.** Text classification is a common task of in-context learning, which may also suffer from a noisy annotation issue. To this end, we verify the effectiveness of the proposed method in text classification. Here, we consider two classification tasks (SST2 [48] and AGNews [67]) with popular label noise types: the symmetric noise and the asymmetric noise [8, 33]. We report the average accuracy with GPT-Neo-2.7B [6] on datasets with the two noise types. More detailed experimental settings are presented in Appendix A.2.

Figure 4 demonstrates that noise annotations barely hurt the performance of ICL when employing the random demonstration selection method [36]. However, the performance of ICL is significantly compromised when utilizing more effective selection methods like TopK [28] and DPP [62]. After integrating our method, both TopK and DPP methods are significantly improved in the inference performance, which indicates the noise robustness of our method in text classification.

**Potential failure cases.** Our approach is built on two assumptions that are naturally satisfied in the real world (See Section 4.2). In this section, we conduct experiments on four generation tasks, including NQ, WebQ, SCIQ, and SQuAD, to determine whether our proposed method remains effective when one of these two assumptions is dissatisfied. The detailed analysis is presented below.

Assumption 1 (Data): clean annotations are the majority in the annotated dataset. Given a dataset with extremely high noise ratios (e.g., 60%, 70%, 80%, 90%), the perplexity ranking of local neighbors may not reflect the correctness of the annotations, as most (even all) neighbors can be wrongly annotated. To explicitly show that, we conduct an experiment to validate the performance of LPR under extremely high noise ratios. The Table 5 below presents the average EM score of the baselines

Table 5: Average test performance of the baselines and our method for four generation tasks on four datasets with extremely high noise ratios (e.g., 60%, 70%, 80%, 90%). The results are shown as Naive/+Ours. The bold indicates the improved results by integrating LPR.

| Method | Irrelevant Noise | | | | Relevant Noise | | | |
|--------|------|------|------|------|------|------|------|------|
| | 60% | 70% | 80% | 90% | 60% | 70% | 80% | 90% |
| Random | 15.80/**26.60** | 11.61/**16.97** | 7.98/**11.24** | 4.79/**5.45** | 27.87/**33.25** | 24.67/**28.29** | 22.51/**24.45** | 20.15/**21.20** |
| TopK | 18.08/**28.08** | 14.62/**18.24** | 10.16/**10.96** | 6.25/**7.17** | 29.55/**34.48** | 26.02/**29.23** | 23.28/**25.87** | 21.21/**22.68** |
| DPP | 16.87/**28.61** | 15.10/**18.01** | 9.93/**10.03** | 6.46/**7.18** | 29.51/**35.19** | 25.85/**28.86** | 23.28/**25.27** | 20.83/**21.95** |

Table 6: Average test performance of the baselines and our method using varying large language models (e.g. OPT-1.3B, OPT-2.7B, OPT-6.7B [66]) across various noise types. The results are shown as Naive/+Ours. The bold indicates the improved results by integrating LPR.

| Method | Clean | Irrelevant Noise | | | Relevant Noise | | |
|--------|------|------|------|------|------|------|------|
| | 0% | 20% | 40% | 60% | 20% | 40% | 60% |
| OPT-1.3B | 13.06/**13.22** | 10.48/**10.96** | 8.66/**9.63** | 5.95/**6.41** | 12.21/**12.58** | 11.33/**11.53** | 10.42/**10.81** |
| OPT-2.7B | 15.30/**15.70** | 12.68/**13.23** | 10.53/**11.45** | 7.01/**9.02** | 14.15/**14.73** | 13.21/**14.33** | 11.86/**12.85** |
| OPT-6.7B | 23.46/**24.03** | 17.26/**21.31** | 11.32/**17.29** | 7.68/**12.91** | 20.16/**22.40** | 17.58/**20.22** | 14.95/**17.52** |

and our method. We use Llama2-7B [49] as the LLM throughout our experiments. The results show that the improvements of our approach decrease as the noise ratios increase. For example, when the irrelevant label noise ratio increases from 60% to 90%, the improvement of our method for the TopK method decreases from 10.26 to 0.92.

Assumption 2 (Model): examples that are semantically similar share the same level of inherent perplexity. The model affects the the performance of LPR through the concept of inherent perplexity. This assumption cannot hold if the model is not capable of precisely measuring the semantic distance between examples. In this case, the local neighbors may not share the same level of inherent perplexity so that we cannot compare the Matching Perplexity. To validate this, we conduct experiments with language models with various sizes, including OPT-1.3B, OPT-2.7B and OPT-6.7B [66]. The results in Table 6 reveal that the performance of LPR decreases as the parameter size of language models decreases. For instance, for 60% irrelevant noise, the improvement of our method decreases from 5.23 to 0.46 when the parameter size of the language model decreases from 6.7B to 1.3B.

# 7 Conclusion

In this paper, we introduce Local Perplexity Ranking (**LPR**), a general strategy that can universally enhance the noise robustness of in-context learning on generation tasks. To the best of our knowledge, this work is the first to analyze the noisy annotations in ICL for text generation. Our key idea is to decouple the matching perplexity by performing the ranking among the neighbors in semantic space. In particular, we replace each low-ranked candidate with its nearest neighbor that is highly ranked. Extensive experiments demonstrate that LPR can improve the noise robustness of existing demonstration selection methods in ICL across various noise types. Our approach is easy to use in practice, as it is insensitive to the hyperparameters and does not introduce heavy computational cost. **Limitations.** LPR is suboptimal in cases of high noise rates due to the assumption that clean annotations are the majority in the dataset. In addition, we do not provide a theoretical analysis to show how noisy annotations affect ICL, which will be an interesting direction for future research.

# 8 Acknowledgements

This research is supported by the Shenzhen Fundamental Research Program (Grant No. JCYJ20230807091809020). Feipeng Zhang is supported by the National Natural Science Foundation of China (Grant No. 72171192) and the Youth Innovation Team of Shaanxi Universities. Jun Shu is supported in part by the National Natural Science Foundation of China (Grant No. 12326606). Feng Zheng is supported in part by the National Natural Science Foundation of China (Grant No. 62122035). We gratefully acknowledge the support of the Center for Computational Science and Engineering at the Southern University of Science and Technology for our research.

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

# A Appendix

## A.1 Related Work

**In-context learning** In-context learning (ICL) has become a new paradigm for natural language processing (NLP), where LLMs make predictions only based on contexts augmented with a few demonstrations [7, 34–36]. The popularity of ICL also raises growing concerns regarding its instability: given different selected demonstrations, ICL's performance can vary from near state-of-the-art to random [9, 12, 15, 26, 32, 37, 55, 54]. Existing studies show that ICL's performance is highly sensitive to order [30, 60], template [37] and labels [50] of selected demonstrations. For example, on the one hand, some previous studies show that flip classification of demonstration can significantly hurt ICL performance on classification tasks [51, 60]. On the other hand, many researches show that ICL is fairly robust to noisy demonstrations [32, 37, 55, 54]. However, the existing studies only focus on classification tasks and the research of generation tasks is limited. We expand the previous finding from text classification tasks to generation tasks and find that demonstrations selected from noisy annotations significantly hurt the ICL performance of generation tasks.

In practice, researchers often use crowdsourcing [61, 73] or large language models (LLMs) [58] such as GPT-4 [38] to create input-output pairs for new tasks, which inevitably leads to some mistakes in the annotations. However, the existing demonstration selection methods for generation tasks such as TopK [28] or DPP [62] only consider the input of demonstrations and assume the demonstrations are selected from a completely clean dataset such as [16, 28, 62, 65]. In comparison, we aim to propose a training-free demonstration selection method for generation tasks that can consistently and significantly improve the robustness of the existing methods under noisy annotations.

**Learning with noisy labels** Label noise is common in many real-world datasets, especially generation tasks [2, 64]. The existing approaches to learning with noisy labels can be classified into two types:(1) training noise-robust models with noisy training datasets: designing noise-robust loss function [1, 63, 52, 72] or designing noise-robust model architectures [2, 18, 64] to mitigate label noise. However, this method is not suitable for ICL, which usually hypothesizes that users are unable to apply fine-tuning techniques [68]. (2) detecting noisy labels and reducing their impacts: comparing model predictions with noisy labels [42, 71] or checking the noisy label consensuses of nearby features [73]. Different from the above literature that focuses on classification tasks, we mainly consider a training-free solution to improve noise-robust ICL for generation tasks.

## A.2 Experimental Setting

**Datasets** We conduct experiments on 6 generation tasks, and examples of each dataset are shown in Tables 12 and 13. For open-domain question-answering tasks, we choose the Natural Questions (NQ) dataset [22] and WebQuestions (WebQ) [5]. For reading comprehension tasks, we choose two reading comprehension datasets: Stanford Question Answering (SQuAD) Dataset [46] and Science Questions (SCIQ) dataset [56]. For code generation tasks, we choose Generating Tabular Answers for Multi-Table Question Answering (GeoQuery) Dataset [39] and Natural Language Interface to the Linux Operating System (NL2Bash) dataset [27]. Following previous studies [16, 25, 62], we report Exact Match (EM) for NQ, WebQ, SQuAD and SCIQ, BLEU for NL2Bash and GeoQuery. We collect these dataset from Huggingface. The train sets of these datasets are regarded as examples datasets and the test sets are used to evaluate the performance of ICL. We randomly subsample 20,000 examples from the train set to generate noisy annotations and select demonstrations. We provide a few examples of noisy annotations of each dataset in Tables 14, 15 and 16.

**Baselines** Our model LPR is essentially a data-centric retriever for in-context demonstration selection. We consider both learning-free and other learning-based retrievers as baselines:

1. **Random** randomly selects demonstrations from a example set without repetition [36].
2. **TopK** retrieve demonstration that are semantically-similar to a test query sample [28].
3. **DPP** uses the original BERT embedding as above without fine-tuning, and adopts MAP inference for subset retrieval [62].

**Experiment details** We run our experiments on NVIDIA L40 GPU. We adopt a large portion of the code from the OpenICL repository [59, 60]. The whole experiment around one week on 8 GPUs and each experiment around one hour on a single GPU.

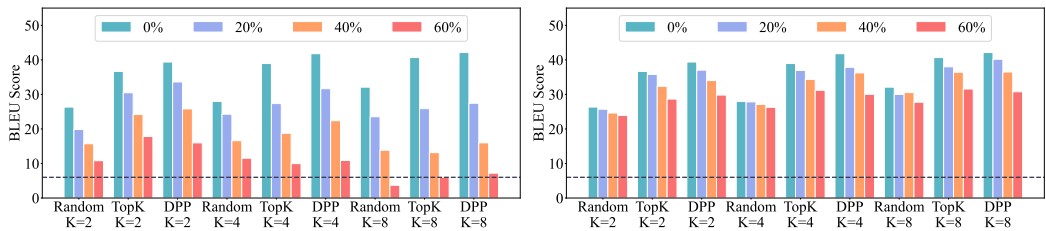

Figure 5: Average results of ICL with noisy annotations in various generation tasks across different demonstration settings. Both the two types of noises significantly deteriorate the performance of in-context learning on code generation tasks. The black line denotes zero-shot performance.

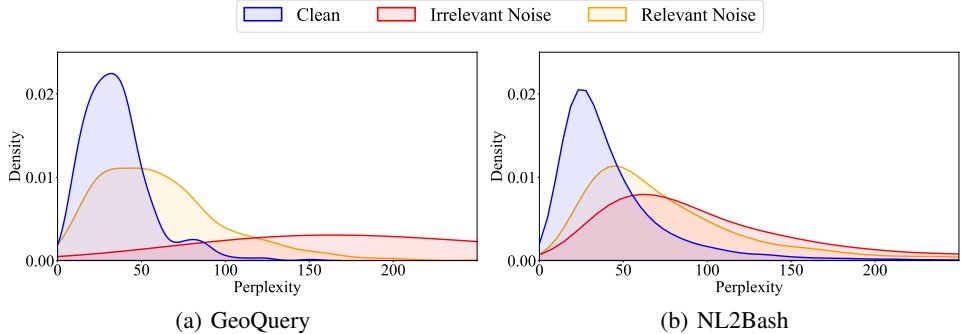

(a) GeoQuery      (b) NL2Bash

Figure 6: The distribution of perplexity of Llama2-7B [49] on clean and noisy annotations. Examples with noisy annotations indeed obtain higher perplexity than those with clean annotations.

**Transfer to classification tasks** Inspired by the idea implemented in above assumption, we assume that examples that are semantically similar share the similar task, indicating they should belong to same classification. We don't need to calculate the perplexity of input-output pair and only identify whether the classification of candidate demonstration is same with its local neighbors or not. Similar to generation tasks, we replace the noisy candidates with their nearest neighbors that are more likely to be clean. We investigate whether our local-based method can transfer across to classification tasks.

## A.3 More empirical results

**Empirical study of noisy ICL in text generation** In this section, we provide the detailed results of GeoQuery and NL2Bash. Following existing studies [25, 62], we adopt BLEU score [41] to evaluate ICL performance on code generation tasks. Figure 5 shows that both the two types of noises significantly deteriorate the performance of in-context learning on code generation tasks. This phenomenon motivates us to further investigate the noise-robustness of in-context learning.

**Perplexity deviation of noisy annotations** In Figure 6, we present the perplexity distribution of Llama2-7B [49] on clean and noisy annotations of GeoQuery and NL2Bash datasets. As a complement, we observe that examples selected from noisy annotations set indeed obtain higher perplexity than those collected from clean annotations, which confirms the deviation can also transfer to code generation tasks.

**Analysis of small noise ratios.** In this section, we conduct experiments on datasets with smaller noise ratios (e.g. 5%, 10%, 15%). Figure 7 (a) and (b) present the average EM score on four generation tasks, including NQ, WebQ, SCIQ, and SQuAD. Figure 7 (a) and (b) show that our method can benefit the ICL performance from a small noise rate (e.g. 5%)

**Open Benchmark Evaluation.** Long-form and open-domain QA tasks such as MT-bench [70] and Arena-Hard [24] serve as valuable additions to the current standardized LLM benchmarks. In this section, we conduct experiments on these complex and open tasks to confirm the effectiveness of our method. The results on MT-bench [70] and Arena-Hard [24] are shown in the Figure 7 (c) and (d), which presents the average answer grading (0-10) [70] of baselines and our method. Figure 7 shows

Table 7: Average test performance of Zero-Shot, In-context learning, Chain-of-Thought (COT) and our proposed method across various noise types. The results are shown as Naive/+Ours. The bold indicates the improved results by integrating LPR.

| Method | Clean | Irrelevant Noise | | | Relevant Noise | | |
|---|---|---|---|---|---|---|---|
| | 0% | 20% | 40% | 60% | 20% | 40% | 60% |
| Zero-Shot | 7.46 | | | | | | |
| Zero-Shot-COT | 10.06 | | | | | | |
| Random-ICL/**+Ours** | 27.94/**28.60** | 24.28/**28.11** | 16.61/**25.78** | 11.53/**22.58** | 26.84/**28.27** | 27.11/**28.95** | 26.26/**26.76** |
| TopK-ICL/**+Ours** | 39.94/**38.62** | 27.34/**36.38** | 18.75/**32.15** | 9.96/**23.93** | 38.94/**36.92** | 34.35/**36.39** | 31.19/**33.62** |
| Manual-COT/**+Ours** | 31.91/31.80 | 26.57/**30.62** | 17.95/**26.64** | 15.30/**23.61** | 30.57/**32.06** | 29.01/**31.02** | 27.13/**30.54** |
| Auto-COT/**+Ours** | 45.69/45.44 | 30.51/**40.10** | 20.51/**34.94** | 10.86/**27.32** | 41.38/**42.78** | 35.91/**40.73** | 27.90/**37.10** |

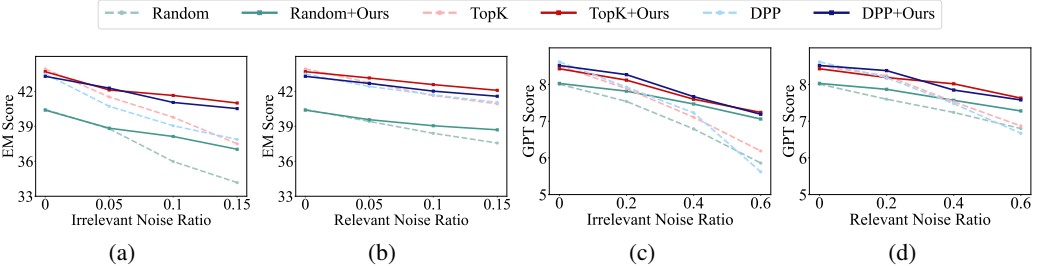

Figure 7: (a) and (b) demonstrate average EM scores of the baselines and our method for four generation tasks on four datasets with smaller noise ratios. (c) and (d) report average GPT-4 score [70] of the baselines and our method for two long-form and open-domin QA datasets.

that our approach significantly improves the efficacy of existing selection methods on long-form question-answering tasks.

**Evaluation on the not-demonstration-selection-based baselines.** Here, we add Zero-Shot baseline, as well as some CoT-related baselines, including Zero-Shot-CoT [20] and Manual-CoT [53], Auto-CoT [69]. Specifically, Manual-CoT [53] and Auto-CoT [69] require to select demonstrations from an annotated examples set. The table below presents the BLEU score of the baselines and our method on the two code generation tasks: Geoquery [39] and NL2Bash [27]. We use Llama2-7B [49] as the LLM throughout our experiments. The results in Table 7 show that our method can outperform Zero-Shot and Zero-Shot-CoT [20], and improve the noise robustness of Manual-CoT [53] and Auto-CoT [69].

**Transfer to similarity score** In LPR, we select reference demonstrations for candidate examples using cosine similarity. While cosine similarity captures some aspects of semantic similarity, it is limited to a single embedding [13]. Another measure of similarity necessitates an accurate characterization of the word levels. One way might be to use larger syntactic substructures of the input as terms with BM25, which is a sparse information retrieval algorithm belonging to a class of TF-IDF measures that view the test input and the candidates as bags of terms and measures relevance as a weighted recall of these terms:

$$\text{BM25}(\boldsymbol{x}_i, \boldsymbol{x}^*) = \sum_i^n W_i R(q_i, \boldsymbol{x}^*)$$

where $q_i$ is each token of $\boldsymbol{x}_i$, $R(q_i, \boldsymbol{x}^*)$ and $W_i$ are the term frequency and inverse document frequency statistics that measure the coverage of a particular term and the relative importance of terms. In this section, we replace the cosine similarity score with the BM25 similarity score to verify the effectiveness of our proposed method.

Our results in Table 8 show that the improvement still holds when BM25 is used as the cluster, confirming the superiority and robustness of our method compared with the naive demonstration selection methods. The above result also demonstrates that examples that are semantically similar in both token space and word space share the same level of inherent perplexity.

**Reordering** Some studies demonstrate that in-context learning is highly sensitive for demonstrations' ordering when using random demonstrations [25, 30, 60]. Specifically, the same randomly sampled demonstrations with different orders can lead to the performance between random guesses and near state-of-the-art. In LPR, we reorder exemplars based on their similarities to the test input in ascending

Table 8: Average results with BM25 as similarity score. The result of The bold indicates the improvement by integrating LPR.

| Dataset | Method | Clean 0% | Irelevant Noise 20% | 40% | 60% | Relevant Noise 20% | 40% | 60% |
|---|---|---|---|---|---|---|---|---|
| NQ | Random | 14.51±0.51 | 10.97±0.29 | 7.37±0.45 | 4.23±0.46 | 12.00±0.65 | 9.67±0.45 | 6.40±1.02 |
| | **+Ours** | **15.15±0.20** | **13.45±0.85** | **10.98±0.47** | **7.51±0.43** | **14.14±0.48** | **12.08±0.53** | **10.12±0.53** |
| | TopK | 20.25±0.10 | 13.95±1.14 | 9.97±1.13 | 5.90±1.08 | 16.21±0.22 | 12.22±0.22 | 8.50±0.28 |
| | **+Ours** | 19.35±0.21 | **17.51±0.08** | **14.44±0.41** | **10.25±0.45** | **17.39±0.45** | **14.38±0.90** | **11.71±0.71** |
| | DPP | 20.35±0.76 | 14.69±0.94 | 9.87±0.49 | 5.97±0.48 | 15.47±1.00 | 11.28±0.42 | 7.89±0.25 |
| | **+Ours** | 19.45±0.97 | **17.29±1.19** | **14.08±0.98** | **10.45±0.68** | **17.07±0.84** | **15.02±0.59** | **12.14±0.75** |
| WebQ | Random | 20.37±0.64 | 15.18±1.06 | 10.39±0.83 | 4.83±0.17 | 18.29±0.43 | 15.92±0.68 | 13.50±0.17 |
| | **+Ours** | **21.18±0.14** | **19.83±0.71** | **16.40±0.28** | **10.89±0.24** | **20.38±0.71** | **18.54±0.48** | **15.92±0.48** |
| | TopK | 30.16±0.58 | 22.52±0.64 | 14.52±0.78 | 8.00±1.12 | 27.19±0.27 | 22.82±0.75 | 18.88±1.09 |
| | **+Ours** | 28.82±0.72 | **26.51±0.39** | **22.03±1.26** | **14.74±0.25** | **27.56±0.20** | **25.08±0.36** | **21.58±0.21** |
| | DPP | 29.40±0.39 | 22.11±0.81 | 13.72±0.27 | 7.33±0.68 | 26.18±1.04 | 21.53±0.61 | 16.80±0.17 |
| | **+Ours** | 29.15±0.21 | **26.30±0.93** | **20.93±1.42** | **13.72±0.57** | **27.83±0.33** | **25.08±0.93** | **20.57±1.27** |
| SQuAD | Random | 56.50±0.57 | 50.00±0.62 | 39.10±0.88 | 26.20±0.79 | 53.90±0.65 | 49.17±0.62 | 42.03±0.79 |
| | **+Ours** | 56.47±0.25 | **54.73±1.10** | **51.53±1.59** | **43.03±1.51** | **54.77±0.76** | **52.83±0.97** | **49.70±0.08** |
| | TopK | 56.97±0.69 | 51.83±1.03 | 42.83±1.68 | 29.10±2.92 | 54.77±0.69 | 49.37±1.37 | 41.37±2.09 |
| | **+Ours** | 56.83±0.19 | **55.60±1.45** | **50.33±0.62** | **40.83±2.82** | **55.70±0.99** | **53.07±0.65** | **48.17±1.92** |
| | DPP | 57.29±0.87 | 50.57±0.33 | 41.63±1.00 | 25.67±2.52 | 56.10±0.59 | 49.57±1.24 | 43.37±0.78 |
| | **+Ours** | 57.20±1.00 | **56.50±0.83** | **52.70±0.86** | **44.73±1.19** | **56.43±1.13** | **53.47±0.81** | **50.57±1.19** |
| SCIQ | Random | 68.15±0.28 | 59.19±1.57 | 44.19±2.89 | 28.21±2.96 | 64.59±1.42 | 58.39±0.16 | 49.54±0.80 |
| | **+Ours** | **69.25±0.86** | **64.14±1.47** | **54.37±1.88** | **37.64±0.58** | **66.49±1.14** | **62.24±0.86** | **54.19±0.82** |
| | TopK | 68.62±1.13 | 59.59±1.28 | 45.77±2.68 | 29.31±1.73 | 64.66±1.34 | 58.54±0.12 | 49.47±0.65 |
| | **+Ours** | **70.11±0.36** | **63.79±2.87** | **57.58±1.52** | **38.90±2.93** | **66.55±2.74** | **60.23±5.44** | **51.95±4.59** |
| | DPP | 67.29±0.35 | 57.69±1.83 | 45.34±1.56 | 28.50±1.78 | 64.88±0.43 | 58.91±0.64 | 50.00±0.85 |
| | **+Ours** | **69.78±1.00** | **64.94±1.42** | **55.34±2.12** | **41.21±1.52** | **67.64±0.86** | **63.85±2.05** | **56.43±2.50** |
| GeoQuery | Random | 27.97±0.99 | 23.18±0.62 | 17.44±1.56 | 14.10±0.74 | 26.48±0.17 | 26.13±0.05 | 26.25±0.40 |
| | **+Ours** | **29.99±0.50** | **29.35±0.26** | **25.69±0.91** | **25.11±0.64** | **29.77±0.35** | **28.09±0.50** | **26.80±0.55** |
| | TopK | 44.17±0.09 | 27.28±2.65 | 17.49±2.05 | 9.96±3.08 | 41.31±0.46 | 38.48±0.63 | 34.90±0.69 |
| | **+Ours** | 43.06±0.60 | **41.61±1.00** | **41.19±1.42** | **32.76±0.45** | **40.99±38.71** | **38.87±0.49** | **36.26±0.03** |
| | DPP | 45.81±0.71 | 31.79±5.93 | 21.54±3.36 | 10.61±0.15 | 42.97±1.96 | 39.91±0.42 | 33.34±0.53 |
| | **+Ours** | 43.92±3.44 | **41.32±3.55** | **38.37±4.19** | **26.78±3.32** | **41.70±1.22** | **39.79±2.13** | **35.34±2.10** |
| NL2Bash | Random | 27.91±0.37 | 25.37±0.21 | 15.77±0.91 | 8.95±0.65 | 27.20±1.06 | 28.09±0.51 | 26.27±0.56 |
| | **+Ours** | **29.15±0.21** | **26.30±0.93** | **20.93±1.42** | **13.72±0.57** | **28.83±0.33** | **28.08±0.93** | **27.57±1.27** |
| | TopK | 35.71±0.42 | 27.40±0.26 | 20.00±0.62 | 9.95±0.68 | 32.57±0.13 | 30.21±0.08 | 27.48±0.35 |
| | **+Ours** | 32.42±0.26 | **29.85±2.99** | **30.10±2.11** | **23.67±1.02** | **31.18±38.71** | **31.03±3.80** | **28.84±2.48** |
| | DPP | 37.77±0.02 | 31.52±0.12 | 23.23±0.34 | 11.16±2.14 | 32.74±0.29 | 32.56±0.61 | 26.72±1.58 |
| | **+Ours** | 36.69±3.30 | **32.63±3.32** | **29.10±4.10** | **23.56±2.65** | **33.18±2.51** | 32.19±3.46 | **28.65±1.80** |

order after the process of our method, in accordance with common practices [23, 28, 45, 47, 62]. Here we compare our method with reordering and without reordering to explore the effect of reordering on example-specific demonstrations retrieved by our method.

Across our experiments, Table 9 shows our method without the reordering process still improves the existing demonstration selection methods across various types of noise. The above results indicate that local perplexity ranking rather than the reordering process is crucial for the success of noise-robust ICL. Additionally, we believe high-quality demonstrations are less sensitive to the ordering and stabilize in-context learning, which is consistent with the previous work [16].

**Various demonstration sizes** To verify the effectiveness of our proposed method, we also present the ICL performance with our method across various set sizes $K$. Concretely, Tables 10 and 11 report the number of demonstrations to be 2 and 8 and show our method effectively mitigates the issue of noisy annotation in various demonstration selection methods across various demonstration sizes.

Table 9: Average results without reordering process. The result of The bold indicates the improvement by integrating LPR.

| Dataset | Method | Clean | Irelevant Noise | | | Relevant Noise | | |
|---|---|---|---|---|---|---|---|---|
| | | 0% | 20% | 40% | 60% | 20% | 40% | 60% |
| NQ | Random | 14.51±0.51 | 10.97±0.29 | 7.37±0.45 | 4.23±0.46 | 12.00±0.65 | 9.67±0.45 | 6.40±1.02 |
| | **+Ours** | **15.35±0.83** | **14.58±0.33** | **12.38±0.09** | **9.24±1.24** | **14.28±0.46** | **12.95±0.91** | **9.93±0.94** |
| | TopK | 20.25±0.10 | 13.95±1.14 | 9.97±1.13 | 5.90±1.08 | 16.21±0.22 | 12.22±0.22 | 8.50±0.28 |
| | **+Ours** | 19.65±0.24 | **16.88±0.40** | **13.21±0.38** | **9.47±0.38** | **17.42±0.36** | **14.58±0.26** | **11.61±0.59** |
| | DPP | 20.35±0.76 | 14.69±0.94 | 9.87±0.49 | 5.97±0.48 | 15.47±1.00 | 11.28±0.42 | 7.89±0.25 |
| | **+Ours** | 18.57±0.24 | **17.45±0.37** | **14.48±0.85** | **11.44±0.29** | **17.75±0.29** | **15.45±0.70** | **12.18±0.87** |
| WebQ | Random | 20.37±0.64 | 15.18±1.06 | 10.39±0.83 | 4.83±0.17 | 18.29±0.43 | 15.92±0.68 | 13.50±0.17 |
| | **+Ours** | **22.08±0.31** | **20.38±0.74** | **16.91±0.61** | **12.16±0.21** | **21.64±0.71** | **19.01±0.78** | **17.06±1.35** |
| | TopK | 30.16±0.58 | 22.52±0.64 | 14.52±0.78 | 8.00±1.12 | 27.19±0.27 | 22.82±0.75 | 18.88±1.09 |
| | **+Ours** | 29.69±0.22 | **26.96±0.66** | **22.12±1.08** | **15.98±0.60** | **29.07±0.04** | **27.26±0.40** | **22.33±1.13** |
| | DPP | 29.40±0.39 | 22.11±0.81 | 13.72±0.27 | 7.33±0.68 | 26.18±1.04 | 21.53±0.61 | 16.80±0.17 |
| | **+Ours** | 29.15±0.21 | **26.30±0.93** | **20.93±1.42** | **13.72±0.57** | **27.83±0.33** | **25.08±0.93** | **20.57±1.27** |
| SQuAD | Random | 56.50±0.57 | 50.00±0.62 | 39.10±0.88 | 26.20±0.79 | 53.90±0.65 | 49.17±0.62 | 42.03±0.79 |
| | **+Ours** | 55.93±0.75 | **54.23±1.11** | **51.67±0.39** | **41.37±0.66** | **55.67±0.52** | **53.13±0.63** | **49.07±0.74** |
| | TopK | 56.97±0.69 | 51.83±1.03 | 42.83±1.68 | 29.10±2.92 | 54.77±0.69 | 49.37±1.37 | 41.37±2.09 |
| | **+Ours** | **57.83±0.97** | **54.87±0.83** | **50.97±0.70** | **39.00±3.12** | **56.40±0.37** | **52.77±0.83** | **47.63±0.94** |
| | DPP | 57.29±0.87 | 50.57±0.33 | 41.63±1.00 | 25.67±2.52 | 56.10±0.59 | 49.57±1.24 | 43.37±0.78 |
| | **+Ours** | **57.47±0.25** | **57.53±0.97** | **52.03±0.39** | **44.00±1.10** | **57.27±0.40** | **55.00±0.22** | **50.27±1.51** |
| SCIQ | Random | 68.15±0.28 | 59.19±1.57 | 44.19±2.89 | 28.21±2.96 | 64.59±1.42 | 58.39±0.16 | 49.54±0.80 |
| | **+Ours** | **68.56±1.17** | **64.88±1.22** | **54.94±1.00** | **40.63±2.62** | **66.67±1.34** | **62.41±0.24** | **54.03±1.69** |
| | TopK | 68.62±1.13 | 59.59±1.28 | 45.77±2.68 | 29.31±1.73 | 64.66±1.34 | 58.54±0.12 | 49.47±0.65 |
| | **+Ours** | **70.00±0.25** | **66.26±0.35** | **56.32±1.90** | **41.03±1.89** | **68.19±0.13** | **63.27±0.75** | **55.17±2.12** |
| | DPP | 67.29±0.35 | 57.69±1.83 | 45.34±1.56 | 28.50±1.78 | 64.88±0.43 | 58.91±0.64 | 50.00±0.85 |
| | **+Ours** | **70.00±0.61** | **66.26±1.27** | **56.03±2.04** | **43.44±2.54** | **68.70±0.21** | **63.22±1.84** | **54.77±2.47** |
| GeoQuery | Random | 27.97±0.99 | 23.18±0.62 | 17.44±1.56 | 14.10±0.74 | 26.48±0.17 | 26.13±0.05 | 26.25±0.40 |
| | **+Ours** | **28.58±0.59** | **28.60±0.03** | **28.89±1.77** | **22.61±0.53** | **27.80±0.27** | **28.45±0.32** | **26.86±0.76** |
| | TopK | 44.17±0.09 | 27.28±2.65 | 17.49±2.05 | 9.96±3.08 | 41.31±0.46 | 38.48±0.63 | 34.90±0.69 |
| | **+Ours** | **45.63±0.11** | **43.62±0.70** | **35.05±2.86** | **26.03±4.93** | **42.74±0.45** | **39.81±0.80** | **35.75±0.03** |
| | DPP | 45.81±0.71 | 31.79±5.93 | 21.54±3.36 | 10.61±0.15 | 42.97±1.96 | 39.91±0.42 | 33.34±0.53 |
| | **+Ours** | 44.73±0.56 | **45.10±0.50** | **40.26±1.06** | **32.54±1.25** | 41.64±0.64 | **40.78±0.89** | **35.09±0.79** |
| NL2Bash | Random | 27.91±0.37 | 25.37±0.21 | 15.77±0.91 | 8.95±0.65 | 27.20±1.06 | 28.09±0.51 | 26.27±0.56 |
| | **+Ours** | 25.54±2.19 | 25.02±1.25 | **23.05±2.22** | **21.28±2.12** | **27.63±0.58** | 24.21±0.66 | 24.09±0.38 |
| | TopK | 35.71±0.42 | 27.40±0.26 | 20.00±0.62 | 9.95±0.68 | 32.57±0.13 | 30.21±0.08 | 27.48±0.35 |
| | **+Ours** | 32.91±0.21 | **31.33±0.50** | **29.83±0.31** | **22.20±0.95** | 31.39±0.74 | **31.14±0.46** | **29.09±1.77** |
| | DPP | 37.77±0.02 | 31.52±0.12 | 23.23±0.34 | 11.16±2.14 | 32.74±0.29 | 32.56±0.61 | 26.72±1.58 |
| | **+Ours** | **38.37±0.32** | **31.81±1.08** | **24.27±2.14** | **13.09±0.05** | **34.43±0.91** | 32.32±1.94 | **28.76±0.88** |

Table 10: Average in-context learning performance with 2 demonstrations on 6 datasets across various types of noisy annotation (over 3 runs). The bold indicates the improved results by integrating LPR.

| Dataset | Method | Clean 0% | Irrelevant Noise 20% | 40% | 60% | Relevant Noise 20% | 40% | 60% |
|---|---|---|---|---|---|---|---|---|
| NQ | Random | 11.70±0.49 | 9.27±0.75 | 6.70±0.81 | 4.57±0.34 | 11.04±0.27 | 8.83±0.54 | 5.80±0.45 |
| | **+Ours** | **12.01±0.71** | **11.38±0.92** | **10.71±0.35** | **8.27±0.75** | **11.58±0.78** | **10.38±0.87** | **8.90±0.51** |
| | TopK | 14.61±0.49 | 12.18±0.05 | 9.53±0.53 | 6.42±0.38 | 11.96±0.62 | 8.89±0.21 | 7.14±0.21 |
| | **+Ours** | 14.28±0.31 | **12.84±0.31** | **11.44±0.54** | **9.31±1.13** | **12.98±0.31** | **11.84±0.46** | **9.77±0.66** |
| | DPP | 15.48±0.26 | 12.03±0.12 | 9.03±0.31 | 6.34±0.49 | 12.21±0.21 | 9.17±0.54 | 6.87±0.36 |
| | **+Ours** | 14.68±0.61 | **13.91±0.65** | **12.11±1.28** | **10.04±1.26** | **14.44±0.50** | **12.41±0.33** | **10.33±0.45** |
| WebQ | Random | 16.06±0.74 | 12.95±0.20 | 9.89±1.38 | 6.65±0.74 | 14.32±0.34 | 13.30±1.45 | 11.74±0.76 |
| | **+Ours** | **16.28±0.65** | **16.20±0.30** | **13.36±0.33** | **14.41±1.22** | **16.58±0.34** | **15.21±0.51** | **13.89±0.43** |
| | TopK | 20.04±0.13 | 16.33±0.17 | 11.88±0.07 | 8.43±0.45 | 17.59±1.41 | 14.82±0.16 | 11.52±0.65 |
| | **+Ours** | **20.50±0.52** | **19.44±0.59** | **17.34±0.95** | **14.41±1.22** | **20.51±0.45** | **18.34±0.84** | **15.87±0.67** |
| | DPP | 22.68±0.61 | 18.10±0.86 | 13.12±0.44 | 8.66±0.51 | 19.60±0.17 | 17.32±0.57 | 13.97±0.63 |
| | **+Ours** | 22.20±0.74 | **20.43±0.40** | **17.96±0.78** | **13.91±0.76** | **21.67±0.35** | **20.79±0.83** | **16.89±0.49** |
| SQuAD | Random | 45.07±0.37 | 41.13±1.03 | 34.77±0.39 | 27.60±1.40 | 43.27±0.59 | 38.83±1.21 | 35.63±0.62 |
| | **+Ours** | 44.57±0.48 | **42.43±1.03** | **42.60±1.02** | **37.40±0.64** | **46.10±1.98** | **42.70±0.65** | **40.03±0.40** |
| | TopK | 45.13±0.76 | 40.57±0.94 | 36.00±0.51 | 29.17±1.33 | 41.73±0.59 | 41.73±0.59 | 35.57±1.26 |
| | **+Ours** | 45.20±0.83 | **44.37±0.65** | **41.17±0.58** | **35.53±1.71** | **44.07±0.45** | **42.47±1.14** | **39.67±1.36** |
| | DPP | 46.23±1.58 | 41.67±1.72 | 35.43±1.76 | 27.90±0.75 | 43.33±0.57 | 40.20±0.50 | 37.23±0.60 |
| | **+Ours** | **46.67±0.59** | **44.53±0.68** | **42.17±1.03** | **37.10±0.41** | **44.77±0.29** | **43.23±0.92** | **41.20±0.22** |
| SCIQ | Random | 66.48±0.34 | 62.01±0.96 | 51.03±2.33 | 40.36±1.94 | 64.65±0.92 | 59.48±0.75 | 56.89±0.25 |
| | **+Ours** | 65.17±0.56 | **62.64±0.69** | **57.59±1.01** | **48.79±1.60** | 64.08±0.80 | **61.61±0.33** | **56.26±1.92** |
| | TopK | 65.17±0.49 | 58.50±1.10 | 50.29±1.06 | 40.54±1.68 | 62.76±1.36 | 58.27±0.73 | 54.54±0.86 |
| | **+Ours** | **67.04±0.50** | **64.60±0.63** | **57.81±1.69** | **49.88±1.79** | **65.63±0.80** | **61.26±0.80** | **55.46±3.17** |
| | DPP | 67.33±0.74 | 61.37±0.98 | 51.49±1.06 | 41.26±1.75 | 62.53±1.06 | 58.16±0.70 | 53.79±0.51 |
| | **+Ours** | 67.24±0.92 | **64.48±1.60** | **59.37±0.84** | **50.57±2.34** | **66.95±2.41** | **62.12±1.18** | **56.84±2.00** |
| GeoQuery | Random | 24.11±1.06 | 18.22±0.87 | 12.35±0.35 | 7.36±0.51 | 24.55±0.42 | 21.55±0.46 | 20.40±0.41 |
| | **+Ours** | 23.67±0.98 | **22.23±0.27** | **19.99±0.17** | **16.29±0.96** | 22.84±0.68 | **22.78±0.22** | **22.34±1.24** |
| | TopK | 41.48±0.41 | 32.11±0.69 | 26.11±1.92 | 18.57±3.32 | 40.08±1.57 | 36.97±1.29 | 33.27±1.88 |
| | **+Ours** | 41.1±0.43 | **40.82±0.70** | **41.51±0.76** | **37.08±0.55** | 38.72±0.92 | **36.61±1.10** | **35.16±0.56** |
| | TopK | 43.63±0.79 | 35.91±3.46 | 25.77±1.34 | 14.66±0.25 | 39.11±1.53 | 35.88±0.98 | 32.01±2.06 |
| | DPP | 41.97±0.05 | **40.67±1.25** | **41.01±0.26** | **36.42±0.12** | 38.13±0.33 | 35.11±0.50 | **33.66±0.05** |
| NL2Bash | Random | 28.56±0.89 | 21.45±2.64 | 19.07±0.75 | 14.25±2.48 | 26.87±0.82 | 25.69±0.26 | 24.47±0.73 |
| | **+Ours** | 26.35±0.20 | **24.37±0.32** | **25.44±1.13** | **20.73±0.29** | 26.22±0.25 | **26.54±0.28** | **26.10±1.75** |
| | TopK | 31.83±0.10 | 28.85±0.68 | 22.3±2.01 | 17.08±3.47 | 31.51±0.82 | 27.73±0.36 | 24.04±0.91 |
| | **+Ours** | **35.10±0.06** | **34.11±0.51** | **30.73±0.47** | **26.04±1.00** | **34.02±0.41** | **30.22±0.54** | **27.12±0.81** |
| | DPP | 35.13±1.07 | 31.31±0.85 | 25.84±0.92 | 17.28±0.72 | 34.95±0.43 | 32.14±0.59 | 27.61±0.85 |
| | **+Our** | 33.79±0.33 | 30.84±0.63 | **30.34±1.02** | **26.41±1.58** | 32.88±1.34 | 31.40±0.59 | **28.82±0.30** |

Table 11: Average in-context learning performance with 8 demonstrations on 6 datasets across various types of noisy annotation (over 3 runs). The bold indicates the improved results by integrating LPR.

| Dataset | Method | Clean 0% | Irelevant Noise 20% | 40% | 60% | Relevant Noise 20% | 40% | 60% |
|---|---|---|---|---|---|---|---|---|
| NQ | Random | 16.25±0.95 | 11.62±0.24 | 6.15±0.51 | 3.17±0.17 | 12.72±0.44 | 9.37±0.24 | 6.17±0.52 |
| | **+Ours** | **16.55±0.47** | **14.08±0.61** | **11.88±0.69** | **8.64±0.70** | **14.58±0.42** | **13.31±0.64** | **9.74±0.74** |
| | TopK | 21.09±0.42 | 14.91±1.26 | 8.57±0.40 | 5.47±0.21 | 17.98±0.34 | 12.71±1.02 | 8.87±0.85 |
| | **+Ours** | 20.65±0.09 | **16.65±0.25** | **12.41±0.65** | **8.14±1.33** | 17.85±0.47 | **15.45±0.82** | **12.01±0.65** |
| | DPP | 19.65±0.31 | 13.34±0.84 | 8.64±0.72 | 5.30±0.00 | 16.31±0.51 | 12.48±1.07 | 8.43±0.48 |
| | **+Ours** | 18.68±0.29 | **16.62±0.45** | **13.71±0.57** | **9.54±1.14** | **17.49±0.41** | **15.82±0.17** | **12.22±1.21** |
| WebQ | Random | 22.70±0.55 | 15.62±0.34 | 8.33±0.31 | 3.22±0.37 | 19.91±0.32 | 17.00±0.73 | 13.52±0.75 |
| | **+Ours** | **22.93±0.47** | **20.62±0.60** | **16.18±0.89** | **9.71±0.23** | **22.36±0.27** | **20.52±0.28** | **17.85±1.05** |
| | TopK | 33.52±0.67 | 22.49±0.95 | 12.51±0.92 | 6.50±0.57 | 28.69±0.69 | 24.17±0.31 | 19.50±0.56 |
| | **+Ours** | 31.64±0.10 | **26.91±0.37** | **19.52±1.29** | **12.71±1.50** | **29.29±0.48** | **26.32±1.39** | **21.86±0.60** |
| | DPP | 31.49±0.27 | 22.66±0.95 | 12.51±0.48 | 5.27±1.43 | 27.64±0.64 | 22.90±0.52 | 17.82±0.00 |
| | **+Ours** | 30.39±0.10 | **26.00±1.01** | **19.08±0.46** | **11.47±0.98** | **28.74±0.43** | **26.43±1.53** | **21.97±1.33** |
| SQuAD | Random | 58.70±0.59 | 46.63±1.20 | 27.80±1.42 | 11.03±0.62 | 54.37±0.66 | 46.57±1.02 | 35.90±1.71 |
| | **+Ours** | 57.73±0.79 | **56.87±0.47** | **48.50±0.86** | **33.00±1.31** | **57.70±0.65** | **53.93±0.33** | **47.57±0.90** |
| | TopK | 58.97±0.42 | 49.80±1.44 | 34.87±1.68 | 15.53±2.23 | 56.63±0.80 | 49.60±0.78 | 36.37±1.16 |
| | **+Ours** | 58.33±0.29 | **55.60±0.16** | **48.30±0.08** | **30.27±2.08** | **58.60±1.56** | **55.13±1.33** | **44.40±0.86** |
| | DPP | 56.93±0.34 | 49.63±1.35 | 33.17±0.45 | 16.50±1.31 | 55.63±1.11 | 49.07±0.74 | 36.47±2.09 |
| | **+Ours** | **57.67±0.82** | **56.30±0.14** | **50.53±1.14** | **34.03±2.43** | **57.83±0.25** | **53.87±1.25** | **44.87±2.05** |
| SCIQ | Random | 68.70±0.16 | 52.47±0.59 | 28.46±1.13 | 12.30±3.25 | 63.79±0.64 | 49.82±1.66 | 37.18±1.23 |
| | **+Ours** | **69.54±0.33** | **62.58±1.22** | **43.79±2.46** | **26.49±1.83** | **66.55±1.70** | **58.62±0.65** | **42.13±3.07** |
| | TopK | 68.91±0.22 | 53.73±0.22 | 31.25±1.35 | 12.87±2.46 | 62.13±0.77 | 50.05±1.05 | 35.32±1.45 |
| | **+Ours** | **70.00±0.49** | **65.23±1.78** | **47.53±3.64** | **28.10±4.98** | **67.87±2.00** | **58.39±3.14** | **43.33±4.97** |
| | DPP | 68.33±0.72 | 55.80±0.90 | 34.54±1.92 | 15.29±3.34 | 62.64±1.87 | 52.41±3.41 | 39.65±2.12 |
| | **+Ours** | **68.39±0.43** | **65.14±1.50** | **48.67±1.62** | **29.13±2.44** | **68.04±1.59** | **58.22±2.01** | **46.09±1.70** |
| GeoQuery | Random | 34.03±0.25 | 25.49±1.64 | 13.95±2.4 | 3.02±0.19 | 32.83±0.25 | 30.98±0.16 | 28.72±0.13 |
| | **+Ours** | 32.48±0.46 | **33.36±1.19** | **31.07±1.66** | **23.76±0.92** | 32.42±0.14 | **31.03±2.29** | **29.88±0.84** |
| | TopK | 45.18±0.48 | 22.07±4.26 | 10.12±0.19 | 3.61±1.39 | 42.63±0.56 | 40.43±0.28 | 34.06±0.76 |
| | **+Ours** | 45.08±0.56 | **41.88±0.10** | **27.72±2.40** | **13.09±1.91** | 42.73±0.53 | **41.37±0.22** | **35.53±1.81** |
| | DPP | 46.71±0.29 | 25.01±2.15 | 15.29±0.05 | 8.10±0.95 | 44.75±0.48 | 39.74±0.23 | 33.43±0.65 |
| | **+Ours** | 45.89±0.27 | **46.09±0.57** | **35.01±0.95** | **23.43±2.93** | **44.91±1.02** | **40.70±0.59** | **34.66±0.60** |
| NL2Bash | Random | 30.17±0.54 | 21.57±1.84 | 13.74±2.31 | 4.37±0.62 | 27.18±0.71 | 28.10±0.39 | 26.75±1.08 |
| | **+Ours** | 29.30±0.97 | **28.38±0.19** | **27.51±2.27** | **18.32±1.50** | 26.58±0.37 | **28.45±0.09** | **27.26±0.80** |
| | TopK | 36.17±1.06 | 29.69±0.16 | 16.17±1.05 | 8.50±1.02 | 33.35±2.46 | 32.42±0.77 | 29.08±1.58 |
| | **+Ours** | 35.16±0.03 | **33.32±0.20** | **27.73±0.97** | **17.82±4.71** | 33.14±0.72 | **32.75±0.20** | **29.69±0.39** |
| | DPP | 37.55±0.56 | 29.87±3.04 | 16.65±0.36 | 6.30±1.20 | 34.61±0.46 | 32.24±0.69 | 28.11±0.64 |
| | **+Our** | 36.93±0.89 | **34.65±0.23** | **30.45±1.17** | **25.36±1.00** | 34.57±1.14 | **33.31±1.26** | **31.90±0.74** |

Table 12: All the datasets used in the experiments.

| Task | Dataset | Train Set | Test Set |
|---|---|---|---|
| Open-Domain QA | NQ [22] | 20,000 | 1,000 |
| | WebQ [5] | 1,261 | 1,213 |
| Reading Comprehension | SQuAD [46] | 20,000 | 1,000 |
| | SCIQ [56] | 6,059 | 581 |
| Code Generation | GeoQuery [39] | 530 | 253 |
| | NL2Bash [27] | 5,000 | 606 |

Table 13: Templates of tasks. Placeholders(e.g. <Question> and <Answer>) will be replaced by real questions or answers.

| Dataset | Prompt | Example |
|---|---|---|
| NQ | **Question**: <Question>
**Answer**: <Answer> | **Question**: The bundles of neurons in the cns are called?
**Answer**: Nucleus |
| WebQ | **Question**: <Question>
**Answer**: <Answer> | **Question**: Where are the libyan refugees going?
**Answer**: Tunisia |
| SQuAD | **Support**: <Support>
**Question**: <Question>
**Answer**: <Answer> | **Support**: Among the philosophies that have influenced modern architects and their approach to building design are rationalism, empiricism, structuralism, poststructuralism.
**Question**: Which philosophy followed structuralism?
**Answer**: poststructuralism |
| SCIQ | **Support**: <Support>
**Question**: <Question>
**Answer**: <Answer> | **Support**: Gravity keeps the Moon orbiting Earth. Gravity keeps the planets orbiting the Sun.
**Question**: What keeps the moon orbiting earth?
**Answer**: gravity |
| GeoQuery | **Question**: <Question>
**Answer**: <Answer> | **Question**: which state is Kalamazoo in
**Answer**: SELECT city.statename FROM city WHERE city.cityname=kalamazoo |
| NL2Bash | **Question**: <Question>
**Answer**: <Answer> | **Question**: Add "execute" to the permissions of all directories in the home directory tree
**Answer**: find -type d -exec chmod +x {}; |

Table 14: An illustration of the effect of the label of demonstration, with three different types of input-label mapping of demonstration. The middle lines are demonstrations, and the last line is the model prediction. The model tends to learn the label of the demonstration.

| NQ Test | **Question**: When did computer become widespread in homes and schools?
**Answer**: | |
|---|---|---|
| **Setting** | **In-Context Demonstration** | **Prediction** |
| Clean | **Question**: When did the internet first become available to the public?
**Answer**: 1980s | 1980s |
| Irrelevant | **Question**: When did the internet first become available to the public?
**Answer**: Crude Oil | Crude Oil |
| Relevant | **Question**: When did the internet first become available to the public?
**Answer**: 2010s | 2010s |
| WebQ Test | **Question**: When did computer become widespread in homes and schools?
**Answer**: | |
| **Setting** | **In-Context Demonstration** | **Prediction** |
| Clean | **Question**: When did the internet first become available to the public?
**Answer**: 1980s | 1980s |
| Irrelevant | **Question**: When did the internet first become available to the public?
**Answer**: Crude Oil | Crude Oil |
| Relevant | **Question**: When did the internet first become available to the public?
**Answer**: 2010s | 2010s |

Table 15: An illustration of the effect of the label of demonstration, with three different types of input-label mapping of demonstration. The middle lines are demonstrations, and the last line is the model prediction. The model tends to learn the label of the demonstration.

| | |
|---|---|
| **SQuAD Input** | **Support**: The Super Bowl 50 Host Committee has vowed to be "the most giving Super Bowl ever", and will dedicate 25 percent of all money it raises for philanthropic causes in the Bay Area. The committee created the 50 fund as its philanthropic initiative and focuses on providing grants to aid with youth development, community investment and sustainable environments.
**Question**: What is the name of the fund that focuses on youth, community and sustainable environments?
**Output**: |

| Setting | In-Context Demonstration | Prediction |
|---|---|---|
| Clean | **Support**: UNFPA works in partnership with governments, along with other United Nations agencies, communities, NGOs, foundations and the private sector, to raise awareness and mobilize the support needed to achieve its mission to promote the rights and health of women and young people.
**Question**: With what sort of agencies does UNFPA work?
**Output**: governments | 25 percent |
| Irrelevant | **Support**: Cells are organized into tissues, tissues are organized into organs.
**Question**: What is considered the smallest unit of the organ?
**Output**: Earth | Earth |
| Relevant | **Support**: Cells are organized into tissues, tissues are organized into organs.
**Question**: What is considered the smallest unit of the organ?
**Output**: tissues | 50 fund |

| | |
|---|---|
| **SCIQ Input** | **Support**: All forms of life are built of at least one cell. A cell is the basic unit of the structure and function of living things.
**Question**: What are the smallest structural and functional units of all living organisms?
**Output**: |

| Setting | In-Context Demonstration | Prediction |
|---|---|---|
| Clean | **Support**: Cells are organized into tissues, tissues are organized into organs.
**Question**: What is considered the smallest unit of the organ?
**Output**: Cells | Cells |
| Irrelevant | **Support**: Cells are organized into tissues, tissues are organized into organs.
**Question**: What is considered the smallest unit of the organ?
**Output**: Earth | Earth |
| Relevant | **Support**: Cells are organized into tissues, tissues are organized into organs.
**Question**: What is considered the smallest unit of the organ?
**Output**: tissues | tissues |

Table 16: An illustration of the effect of the label of demonstration, with three different types of input-label mapping of demonstration. The middle lines are demonstrations, and the last line is the model prediction. The model tends to learn the label of the demonstration.

| GeoQuery Test | **Question**: How high is the highest point of Alabama?
**Answer**: |
|---|---|
| **Setting** | **In-Context Demonstration** |
| Clean | **Question**: How high is the highest point in Montana?
**Answer**: SELECT highlow.highest.elevation FROM highlow WHERE highlow.statename='Montana'
**Prediction**: SELECT highlow.highest.elevation FROM highlow WHERE highlow.statename='Alabama' |
| Irrelevant | **Question**: How high is the highest point in Montana?
**Answer**: more than 900 million
**Prediction**: What are the highest point in Alabama |
| Relevant | **Question**: How high is the highest point in Montana?
**Answer**: SELECT city.cityname FROM city WHERE city.statename='Montana'
**Prediction**: SELECT city.cityname FROM city WHERE city.statename='Alabama' |
| NL2Bash Test | **Question**: List all files in the current directory tree larger than 1000 kb
**Answer**: |
| **Setting** | **In-Context Demonstration** |
| Clean | **Question**: Find and show all files in the current directory tree that are exactly 1000 kB.
**Answer**: find . -size 1000k
**Prediction**: find . -size +1000k |
| Irrelevant | **Question**: Find and show all files in the current directory tree that are exactly 2000 kB?
**Answer**: Arizona Department of Water Resources
**Prediction**: 3 files |
| Relevant | **Question**: Find and show all files in the current directory tree that are exactly 2000 kB?
**Answer**: find . -type f -size 2000 -name "*.err"
**Prediction**: find . -type f -size +1000 -name "*.err" |

