# OpenReview forum: "On the Noise Robustness of In-Context Learning for Text Generation"
_NeurIPS.cc/2024/Conference — NeurIPS 2024 poster_

### Official Review · Reviewer_DyDA · 2024-07-01

**Soundness:** 3
**Presentation:** 3
**Contribution:** 3
**Rating:** 7
**Confidence:** 3

**Summary:**

The paper introduces a new method to deal with noisy annotations for in-context learning. The authors suppose that the examples that cause higher perplexity than their neighbors are more likely corrupted than their neighbors. So, the authors suggest replacing the examples, causing suspiciously high perplexity by their neighbors with lower perplexity (the process is formalized by the "Local Perplexity Ranking"  formula). They show experimentally that this method significantly improves the quality of in-context learning for several tasks for several models of size 13B or less.

**Strengths:**

- The paper approaches an important problem with a novel method that significantly improves the quality of LLMs in in-context learning scenarios;
- The main idea of the paper is easy to understand.

**Weaknesses:**

- In Table 2, the authors compare their method with several demonstration selection methods. However, it is still unclear whether **any** demonstration selection method, including LPR, is really the best way to improve the model quality in this situation. What about simpler methods to improve the answer quality, such as a chain of thought? What if it could work better even when coupled with the demonstration of noisy examples? Adding more types of baselines, such as some simple variations of a chain of thoughts and chain of thoughts + noisy examples would make the paper's claims much more solid, but there is no such comparison.
- See the "Questions" section and "Limitations" section.

**Questions:**

Essential questions:
- Why Perplexity = Inherent Perplexity + Matching Perplexity? I didn't find any experimental or theoretical confirmation.
- In Figure 2, there is the perplexity distribution of Llama2-7B on clean and noisy annotations. Does this perplexity distribution change for bigger models, such as Llama-13B? Does it change for smaller models, such as OPT?
- How to choose a good gamma? You wrote in lines 233-234 that "LPR shows robustness to the choice of threshold γ", but I didn't find any experimental confirmation of this point.

Questions about the presentation:
- What model is used for Table 2 results? Is it LLAMA-2 7B?
- What metric is shown in Table 3?
- You wrote that figures 3 (a)-(d) use different gamma values, but which exactly? There is no information in the figures caption.

**Limitations:**

I would suggest adding the lack of any not-demonstration-selection-based baselines (such as chain of thought) and a narrow scope of models to the limitations section. **Or** I would suggest performing corresponding experiments and adding them to the paper if it is possible to do so in a limited time.

---

> ### Author Rebuttal · Authors · 2024-08-06
>
> Thanks for your recognition and the valuable suggestions. Please find our response below.
>
> **1. Evaluation on the not-demonstration-selection-based baselines  [W1, L1]**
>
> Thank you for the great suggestion. Here, we add Zero-Shot baseline, as well as  some CoT-related baselines, including Zero-Shot-CoT [1] and Manual-CoT [4], Auto-CoT [5]. Specifically, Manual-CoT [4] and Auto-CoT [5] require to select demonstrations from an annotated examples set. The table below presents the BLEU score [2, 3] of the baselines and our method on the two code generation tasks: Geoquery and NL2Bash. We use Llama2-7B as the LLM throughout our experiments. The results show that **our method can outperform Zero-Shot and Zero-Shot-CoT**, and **improve the noise robustness of Manual-CoT and Auto-CoT**. We will add the results in the final version.
>
>
> |Type |Clean|Irrelevant|||Relevant|||
> |-|-|-|-|-|-|-|-|
> |**Method**|0%|20%|40%|60%|20%|40%|60%|
> |Zero-Shot|7.46|-|-|-|-|-|-|
> |Zero-Shot-CoT|10.06|-|-|-|-|-|-|
> |Random-ICL|27.94|24.28|16.61|11.53|26.84|27.11|26.26|
> |**+Ours**|**28.60**|**28.11**|**25.78**|**22.58**|**28.27**|**27.95**|**26.76**|
> |TopK-ICL|39.94|27.34|18.75|9.96|38.94|34.35|31.19|
> |**+Ours**|38.62|**36.38**|**32.15**|**23.93**|**36.92**|**35.39**|**33.62**|
> |Manual-CoT|31.91|26.57|17.95|15.30|30.57|29.01|27.13|
> |**+Ours**|31.80|**30.62**|**26.64**|**23.61**|**32.06**|**31.02**|**30.54**|
> |Auto-CoT|45.69|30.51|20.51|10.86|41.38|35.91|27.90|
> |**+Ours**|45.44|**40.10**|**34.94**|**27.32**|**42.78**|**40.73**|**37.10**|
>
> **2. Justification of perplexity disentanglement [Q1]**
>
> Thank you for pointing out the potential confusion due to ambiguous descriptions. We would like to clarify that the disentanglement is conceptual rather than mathematical, which is provided to help the reader understand the justification of LPR. We provide a detailed explanation of the concept below.
>
> * Given a matched input-output pair $\textbf{z}=(\textbf{x}, \textbf{y})$, the $\operatorname{Perplexity}(\textbf{z})$ measures how the model is familiar with the task. In this case, we define **inherent perplexity = $\operatorname{Perplexity}(\textbf{z})$** and **matching perplexity=0**.
> * By replacing the output with a wrong one, we obtain a mismatched input-output pair $\tilde{\textbf{z}}=(\textbf{x}, \tilde{\textbf{y}})$. During the replacement, **inherent perplexity** is not changed as it only depends on the input and its ground-truth output. We define the **matching perplexity** as the increased perplexity caused by the wrong output ($\operatorname{Perplexity}(\tilde{\textbf{z}})-\operatorname{Perplexity}(\textbf{z})$).
>
> Therefore, the disentanglement naturally holds due to their definitions. The concept of **matching perplexity** enables us to compare the correctness of different examples regardless of their inherent perplexity (i.e., task difficulty): a higher matching perplexity indicates that the output is more likely to be incorrect for the input. As the ground-truth is unknown in the dataset, it is non-trivial to explicitly compute the inherent and matching perplexity, which motivates our method. In the final version, we will improve the writing of the perplexity disentanglement to avoid any potential confusion.
>
>
> **3. Results of other LLMs [Q2]**
>
> Great suggestion. To demonstrate that our perplexity analysis is model-agnostic, we add an experiment that presents the perplexity distribution shift of Llama2-13B and OPT-6.7B on clean and noisy annotations. The results are presented in Figures 2 and 3 in the submitted PDF. The results show that examples with noisy annotations exhibit higher perplexity than those with clean annotations for both Llama2-13B and OPT-6.7B, which aligns with our previous findings using the Llama2-7B model.
>
>
> **4. Analysis of threshold $\gamma$ [Q3]**
>
> Sorry for the confusion. The experimental results of various thresholds $\gamma$ are presented on Figure 3 (a) and (b), which report the average test performance across various noise types. Three colors indicate different values of thresholds $\gamma$ (e.g., 25%, 50%, 75%), as presented in the legend. The results show that the performance of LPR is insensitive to the hyperparameter $\gamma$. For example, LPR yields significant improvements of EM score even when $\gamma$=75%. Therefore, **we simply set the threshold $\gamma$= 50% by default**, in our experiments. To avoid any potential confusion, we will improve the legend and caption of Figure 3 in the final version.
>
>
> **5. Concerns of presentation [Q4]**
>
> Thank you for pointing out the issues of presentations. We reply to each question below.
> * We use Llama2-7B as the LLM throughout our experiments, which is described in Subsection 5.1.
> * In Table 3, we report the average Exact Match (EM) score of baselines and LPR using various LLMs.
> * In Figure 3 (a) and (b), we analyze how the hyperparameter $\gamma$ affects the performance of LPR. We present the value of $\gamma$ by the percentages in the legend of Figure 3 (a) and (b) (i.e., 25%, 50%, and 75%), represented by different colors.
>
> As the reviewer suggested, we will improve the presentation in the final version.
>
>
> [1] Kojima T, et al. Large language models are zero-shot reasoners. NeurIPS 2022.
>
> [2] Li X, et al. Unified demonstration retriever for in-context learning. ACL 2023.
>
> [3] Lin X, et al. Nl2bash: A corpus and semantic parser for natural language interface to the Linux operating system. LREC 2018.
>
> [4] Wei J, et al. Chain-of-thought prompting elicits reasoning in large language models. NeurIPS 2022.
>
> [5] Zhang Z, et al. Automatic chain of thought prompting in large language models. ICLR 2023.

---

### Official Review · Reviewer_tEJo · 2024-07-23

**Soundness:** 3
**Presentation:** 4
**Contribution:** 3
**Rating:** 7
**Confidence:** 3

**Summary:**

The paper "On the Noise Robustness of In-Context Learning for Text Generation" investigates how LLMs handle noisy annotations during in-context learning (ICL). The authors propose a method called Local Perplexity Ranking (LPR) that replaces noisy candidates with nearby neighbors that are less noisy. They also explore the impact of noisy labels on ICL performance and compare different selection methods for these examples, such as TopK and DPP. Their proposed method can effectively mitigate the negative effects of noise, offering insights into enhancing the noise robustness of ICL in practical applications.

**Strengths:**

- The paper introduces a novel approach (LPR) to address the issue of noisy annotations in in-context learning for text generation tasks. This is particularly original as it challenges previous assumptions about the robustness of in-context learning to noisy demonstrations in classification tasks.
- The paper is generally well-structured and easy to follow. The definitions and metrics are defined properly. The methodology is described in clear detail.
- The paper shows that LPR does improve the performance of various selection methods, including Random, TopK, and DPP, especially in the presence of noisy annotations. The proposed method addresses an important and practical issue in the field.

**Weaknesses:**

- The absence of code makes it difficult for the broader research community to reproduce the results claimed in the paper or verify the method's effectiveness on the tasks (the authors claim that the code is released in section 4, but I'm not sure where it is).
- The paper did not provide an analysis of the types of errors that LPR makes versus the baseline methods. Such an analysis could provide insights into the strengths and weaknesses of the approach.
- While LPR is compared to baseline selection methods like TopK and DPP, it is not compared to other techniques specifically designed to handle noisy annotations during in-context learning.

**Questions:**

- What are the specific cases where LPR may tend to fail? Does each model's inherent capabilities affect these results?
- What is the specific model version for GPT-4 and the approximate cost for generating relevant noise for each task? What was the specific method (or prompt) used for generating relevant/irrelevant noise?

**Limitations:**

The authors have addressed the potential limitations of this paper. It would be useful if the authors also present examples where their proposed method have failed.

---

> ### Author Rebuttal · Authors · 2024-08-06
>
> Thanks for your recognition and the valuable suggestions. Please find our response below.
>
> **1. The implementation code is missing [W1]**
>
> The code and data have been provided in the **Supplementary Material** of the original submission. Once published, we will upload the code and data to **GitHub** for reproduction.
>
> **2. Analysis of error types [W2]**
>
> We guess your concern is about type I and type II errors that LPR makes versus the baseline methods. However, we primarily focus on text-generation tasks, such as question-answering and code-generation tasks, where type I and type II errors are not applicable. We would be grateful if you could give clear information about the error types here.
>
> **3. Comparison with other methods for noisy ICL [W3]**
>
> Thanks for the suggestion. To the best of our knowledge, this work is the first to explore the noise robustness of in-context learning for text-generation tasks. Indeed, some works focus on the noisy ICL for classification, but they do not propose methods as noisy annotations are not harmful in their analysis. Therefore, our evaluations are conducted to show the improvements of LPR based on existing selection methods. We would be grateful if you could share some related methods we can include in the comparison.
>
> **4. Potential failure modes of LPR [Q1, L1]**
>
> Thank you for the suggestion on potential failure cases for our method. Please find the detailed analysis in the **General Response**.
>
> **5. Details of  noise generation [Q2]**
>
> Thank you for the questions. We respond to these questions point by point below.
> * **Version & expenses**. The model version of GPT-4 we used is GPT-4-0125-preview. The approximate expense is \$20 (2M tokens) per task for open-domain question-answering and code-generation tasks. For the reading comprehension task, the approximate expense is 50 (5M tokens) per task.
> * **Method & Prompt**. For irrelevant noise, we collect multiple irrelevant text-generation datasets and randomly match the answers from the irrelevant datasets as irrelevant noise. Based on the previous works on modeling relevant noise [1, 2, 3], we simulate the relevant noise by generating related yet incorrect outputs using GPT-4-0125-preview. Taking the Natural Questions (NQ) dataset as an example, we generated relevant noises using GPT-4-0125-preview with the following prompt:
>
> > Hello, you are now a data annotation engineer. You will be provided with a question and its correct answer. You need to generate answers that are related but contain obvious errors. Below are two examples.
> >
> > Example 1
> >
> > Question: Where is the liver in the human body located?
> >
> > Correct Answer: The right upper quadrant.
> >
> > Incorrect Answer: The left lower quadrant.
> >
> >
> > Example 2
> >
> > Question: What major cellular event happens during the S phase of interphase?
> >
> > Correct Answer: DNA replication.
> >
> > Incorrect Answer: RNA replication.
> >
> > Question: What is the term given to the energy released into space?
> >
> > Correct Answer: Radiant light energy.
> >
> > Incorrect Answer:
>
> Given the above prompt, the output of GPT-4-0125-preview is "Gravitational potential energy". We will add details of noise generation in the appendix of the final version.
>
> [1] Alexandrov D, et al. Does noise really matter? investigation into the influence of noisy labels on Bert-based question-answering system. International Journal of Semantic Computing, 2024.
>
> [2] Welbl, et al. Crowdsourcing Multiple Choice Science Questions. WNUT 2017.
>
> [3] Wretblad N, et al. Understanding the Effects of Noise in Text-to-SQL: An Examination of the BIRD-Bench Benchmark. ACL 2024.

---

> > ### Comment · Reviewer_tEJo · 2024-08-11
> > **Response to authors**
> >
> > The authors' response resolves most of my issues. Thus, I have changed my score accordingly.

---

> > > ### Author Response · Authors · 2024-08-11
> > > **Thank you for increasing the score**
> > >
> > > We appreciate the valuable suggestions and feedback from the reviewer. We are also glad that most of your concerns have been addressed. Thanks again for increasing the rating!

---

### Official Review · Reviewer_Fz4p · 2024-07-24

**Soundness:** 3
**Presentation:** 3
**Contribution:** 2
**Rating:** 5
**Confidence:** 3

**Summary:**

This paper proposes Local Perplexity Ranking (LPR), a method to improve the robustness of in-context learning for text generation tasks when dealing with noisy annotations. The key contributions are:

- Empirically demonstrating that noisy annotations hurts performance of in-context learning for text generation, unlike for classification tasks.

- Proposing LPR, which ranks candidate demonstrations based on perplexity within local semantic neighborhoods to identify and replace likely noisy examples.

- Experiments showing LPR improves noise robustness in most scenarios across multiple text generation datasets, 2 noise types, and 3 baseline selection methods.

The method is motivated by analyzing perplexity distributions of clean vs noisy examples and decomposing perplexity into inherent and matching components. Overall, LPR provides a simple but effective approach to mitigate issues with noisy demonstrations in in-context learning for text generation.

**Strengths:**

1. Provides clear analysis on the effect of noisy labels to text generation tasks.

2. The explanations in the disentanglement of perplexity justifies the method of LPR.

3. Comprehensive empirical analysis and ablation studies in the appendix.

**Weaknesses:**

1. The author states that the paper's motivation stems from the occurrence of noisy annotations in in-context demonstrations:
> For those candidates, input-label mappings solicited from humans [58, 68] or LLMs [55] can often be noisy, especially in **complex tasks**. This gives rise to the importance of noise-robust ICL, which aims to construct effective demonstrations in the presence of noisy and erroneous labels.

However, the current evaluation of the proposed LPR approach is limited to short-form, closed-domain question-answering tasks using traditional NLP datasets. These datasets typically don't suffer from noisy annotations, as the tasks are relatively simple, and ensuring the correctness of a few (fewer than 10) in-context samples shouldn't be challenging. Including experiments on long-form, open-domain question-answering tasks would better justify the paper's motivation and demonstrate the broader applicability of LPR to more novel tasks.

2. The evaluation experiments on justifying the benefits of LPR are conducted using noisy annotations with noise ratios exceeding 20%, which is likely unrealistic for short-form QA tasks in real-world scenarios. This experimental setting appears overly synthetic.

3. It would be helpful if the author could provide a clear illustration demonstrating how LPM is conducted.

**Questions:**

1. Have the authors conducted experiments on long-form, open-domain question-answering tasks (e.g., MT-Bench)?

2. I observe that the benefits of LPR decrease as the noise level diminishes. In fact, in Table 2, when the labels are clean, the effect of LPR appears negligible, as it doesn't improve upon the baseline 50% of the time. While I understand that LPR is designed to handle noisy labels, have the authors conducted experiments with noise ratios between 0% and 20%? At what threshold does LPR start to show a noticeable effect compared to the baseline?

**Limitations:**

Yes. Would be great to further discuss some possible failure mode of LPR.

---

> ### Author Rebuttal · Authors · 2024-08-06
>
> Thank you for the constructive and elaborate feedback. Please find our response below.
>
> **1. Clarification of motivation and evaluation [W1, Q1]**
>
> There might be some misunderstandings, which are clarified in the following.
>
> * **"The evaluated tasks are simple"**. In this work, we evaluate the effectiveness of our method(LPR) on six datasets of text generation (See Sections 3 and 5), including **Open-Domain Question Answering**: NQ, WebQ; **Reading Comprehension**: SQuAD and SCIQ; **Code Generation**: GeoQuery and NL2Bash. With these tasks ranging from easy to hard, we provide an extensive evaluation for the efficacy of LPR. As suggested by the reviewer, we also add two long-form and open-domain QA tasks, **MT-Bench** [6] and **Arena-Hard** [3].
> * **"Ensuring the correctness of a few (fewer than 10) in-context samples shouldn't be challenging"**. We would like to justify that noisy annotations we considered exist in the large-scale dataset with annotations, instead of simply the selected in-context samples. The in-context samples for each test instance are **unique**, which depend on the selection methods (e.g., Random, TopK, DPP). Thus, it is impractical to ensure the correctness of in-context samples for all test instances by human intervention.
>
> The results on MT-Bench [6] and Arena-Hard [3] are shown in the table below, which presents the average answer grading (0-10) [6] of baselines and our method. The results show that our approach significantly improves the efficacy of existing selection methods on long-form question-answering tasks. We will add the detailed results in the final version.
>
> |Type|Clean|Irrelevant|||Relevant|||
> |-|-|-|-|-|-|-|-|
> |**Method**|0%|20%|40%|60%|20%|40%|60%|
> |Random|8.01|7.54|6.79|5.86|7.60|7.24|6.80|
> |**+Ours**|8.03|**7.82**|**7.47**|**7.06**|**7.87**|**7.57**|**7.28**|
> |TopK|8.51|7.89|7.11|6.19|8.24|7.53|6.87|
> |**+Ours**|8.43|**8.12**|**7.60**|**7.24**|8.19|**8.02**|**7.63**|
> |DPP|8.62 |7.93|7.23|5.62|8.19|7.48|6.67|
> |**+Ours**|8.52|**8.27**|**7.67**|**7.19**|**8.38**|**7.85**|**7.58**|
>
> **2. Analysis of small noise ratios [Q2]**
>
> Thank you for the suggestion. Following the reviewer's advice, we conduct experiments on datasets with smaller noise ratios (e.g. 2%, 5%, 10%, 15%). The table below presents the average EM score on four generation tasks, including NQ, WebQ, SCIQ, and SQuAD. The Table shows that our method can benefit the ICL performance from a small noise rate (e.g. 2%). We will add the results in the final version.
>
> |Type|Clean|Irrelevant||||Relevant||||
> |-|-|-|-|-|-|-|-|-|-|
> |**Method**|0%|2%|5%|10%|15%|2%|5%|10%|15%|
> |Random|40.45|39.39|38.79|35.98|34.17|39.97|39.42|38.40|37.57|
> |**+Ours**|40.39|**39.50**|**38.84**|**38.13**|**37.03**|39.95|**39.57**|**39.05**|**38.70**|
> |TopK|43.93|42.50|41.52|39.78|37.52|43.48|42.75|41.64|40.93|
> |**+Ours**|43.69|**42.91**|**42.12**|**41.65**|**41.00**|43.42|**43.15**|**42.58**|**42.09**|
> |DPP|43.49|42.32|40.72|39.05|37.88|42.92|42.42|41.70|41.06|
> |**+Ours**|43.29|**42.64**|**42.28**|**41.05**|**40.52**|**43.04**|**42.68**|**42.02**|**41.58**|
>
> **3. Noise ratios exceeding 20% are unrealistic [W2]**
>
> Thank you for the good question about the noisy settings of ICL. We have provided more results with noise ratios under 20% in the above. Here, we provide justifications of high noise ratios below:
>
> * **The evaluated tasks are not simple**. As clarified above, we conducted evaluations on various tasks ranging from easy to hard. In hard tasks, e.g., code generation, it is reasonable to utilize a large-scale dataset collected from forums, where the noise rate exceeds 20%.
> * **It is common practice to explore the issue of noisy annotations with high ratios in the literature**. Recent work shows that numerous QA datasets annotated by humans or LLMs exhibit a high proportion of corrupted outputs ranging from 8% to 38.5% [4]. Jindal et al. (2019) [2] and Xu et al. (2024) [5] introduced noisy labels, ranging from 20% to 60%, into text classification datasets. Alexandrov et al. (2024) [1] define four levels of injected noises from 20% to 80% for the finetuning of Bert. Thus, it is necessary to justify the benefits of LPR with noise ratios exceeding 20%.
> * **There is a lack of real-world benchmarks for noisy ICL research**. While the conventional wisdom in the ICL community is that noisy annotations do not affect the efficacy of ICL in classification tasks [2, 5], this work is the first to show the harmfulness of noisy annotations in ICL. Thus, there is no existing real-world benchmark for noisy ICL, which will be urgently needed after the publication of this work.
>
> In summary, conducting evaluation on synthetic datasets with high noise ratios is reasonable due to the task difficulty, common practices, and the lack of real-world benchmarks. We hope this work can attract the attention of researchers on the label quality of ICL datasets, such as collecting real-world benchmarks.
>
> **4. Clear illustration of LPR [W3]**
>
> Thank you for the great suggestion.  We provide a clear illustration to express the framework of LPR, as shown in Figure 1 in the submitted PDF. We will add the illustration in the revised version.
>
> **5. Potential failure of LPR [L1]**
>
> Thank you for the suggestion on potential failure cases for our method. Please find the detailed analysis in the General Response.
>
> [1]Alexandrov D, et al. Does noise really matter? investigation into the influence of noisy labels on Bert-based question-answering system. IJSC 2024.
>
> [2]Jindal I, et al. An effective label noise model for DNN text classification. NAACL 2019.
>
> [3]Li T, et al. From Crowdsourced Data to High-Quality Benchmarks: Arena-Hard and BenchBuilder Pipeline. arXiv 2024.
>
> [4]Song H, et al. Learning from noisy labels with deep neural networks: A survey. TNNLS 2022.
>
> [5]Xu P, et al. Noisy Multi-Label Text Classification via Instance-Label Pair Correction. NAACL 2024.
>
> [6]Zhang L, et al. Judging llm-as-a-judge with mt-bench and chatbot arena. NeurIPS 2023.

---

> > ### Comment · Reviewer_Fz4p · 2024-08-08
> > **Response to the authors**
> >
> > I appreciate the author's responses and the experiments conducted on MT-Bench and Arena-Hard, as well as the additional experiments with small noise ratios. I have raised my score accordingly.

---

> > > ### Author Response · Authors · 2024-08-08
> > > **Thank you for raising the score**
> > >
> > > Thank you for checking our rebuttal and raising your score. We will incorporate the new results and explanations into the final version appropriately.  Sincerely thanks for your valuable time on this paper!

---

### Author Rebuttal · Authors · 2024-08-07

# **General Response**

We thank All the reviewers for their time and valuable comments. We are glad that reviewers find this work focuses on an **important** and **practical** problem (tEJo, DyDA) with **clear** analysis (Fz4p). We are also encouraged that reviewers find that the method is **novel/original** (tEJo, DyDA), **simple but effective** (Fz4p), and the experiments are **comprehensive** (Fz4p) with **significant** improvements (DyDA). Besides,  reviewers recognize that this paper is **clear** (tEJo), **well-structured** (tEJo), and **easy to understand** (tEJo, DyDA). We provide point-by-point responses to all reviewers’ comments and concerns.

In the following responses, we have addressed the reviewers' comments and concerns point by point and are willing to discuss any concerns you may have.

## Potential failure cases

We note that two reviewers are interested in the potential failure cases of our method. Here, we provide a detailed analysis of this concern.

In general, our proposed LPR may fail to help when the two natural assumptions of our method (See Section 4.2) are dissatisfied. The detailed analysis is presented below.
* **Assumption 1 (Data): clean annotations are the majority in the annotated dataset**. Given a dataset with extremely high noise ratios (e.g., 70%, 80%, 90%), the perplexity ranking of local neighbors may not reflect the correctness of the annotations, as most (even all) neighbors can be wrongly annotated. To explicitly show that, we conduct an experiment to validate the performance of LPR under extremely high noise ratios. The table below presents the average EM score of the baselines and our method for four generation tasks, including NQ, WebQ, SCIQ, and SQuAD. The results show that **the improvements of our approach decrease as the noise ratios increase**. For example, when the irrelevant label noise ratio increases from 60% to 90%, the improvement of our method for the TopK method decreases from 10.26 to 0.92.

|Type|Irrelevant||||Relevant||||
|-|-|-|-|-|-|-|-|-|
|Method|60%|70%|80%|90%|60%|70%|80%|90%|
|Random|15.80|11.61|7.98|4.79|27.87|32.16|30.75|24.55|
|**+Ours**|**26.60**|**16.97**|**11.24**|**5.45**|**33.25**|**34.08**|**31.03**|**25.53**|
|TopK|18.08|14.62|10.16|6.25|29.55|31.29|27.55|25.60|
|**+Ours**|**28.34**|**18.24**|**10.96**|**7.17**|**34.48**|**32.78**|**28.79**|**27.35**|
|DPP|16.87|15.10|9.93|6.46|29.51|30.82|28.29|25.33|
|**+Ours**|**28.61**|**18.01**|**10.03**|**7.18**|**35.19**|**32.85**|**30.97**|**27.37**|

* **Assumption 2 (Model): examples that are semantically similar share the same level of inherent perplexity.** The model capability affects the performance of LPR through the concept of inherent perplexity. This assumption cannot hold if the model is not capable of precisely measuring the semantic distance between examples. In this case, the local neighbors may not share the same level of inherent perplexity so that we cannot compare the Matching Perplexity. To validate this, we conduct experiments with language models with various sizes, including OPT-1.3B, OPT-2.7B and OPT-6.7B. The results reveal that **the performance of LPR decreases as the parameter size of language models decreases**. For instance, for 60% irrelevant noise, the improvement of our method decreases from 5.23 to 0.46 when the parameter size of the language model decreases from 6.7B to 1.3B.


|Type|Clean|Irrelevant|||Relevant|||
|-|-|-|-|-|-|-|-|
|Model|0%|20%|40%|60%|20%|40%|60%|
||Naive/**+Ours**|||||||
|OPT-1.3B|13.06/13.22|10.48/**10.96**|8.66/**9.63**|5.95/**6.41**|12.21/**12.58**|11.33/**11.53**|10.42/**10.81**|
|OPT-2.7B|15.30/**15.70**|12.68/**13.23**|10.53/**11.45**|7.01/**9.20** |14.15/**14.73**|13.21/**14.33**|11.86/**12.85**|
|OPT-6.7B    |23.46/**24.03**|17.26/**21.31**|11.32/**17.29**|7.68/**12.91**|20.16/**22.40**|17.58/**20.22**|14.95/**17.52**|

In summary, our method may fail to improve the noise robustness when the noise ratio is extremely high or the model capability is too weak. We will present the analysis in the final version.

---

### Decision · Program_Chairs · 2024-09-25

**Decision:**

Accept (poster)

**Comment:**

This paper introduces a method to improve the robustness of in-context learning for text generation tasks. They identified the importance of using clean examples as demonstrations for text generation tasks. Authors also addressed reviewers' suggestions well by adding experiments on real-world open domain QA results and comparisons with chain-of-thought methods. Overall, this paper presents solid experimental results and analysis.